# ResFields: Residual Neural Fields for Spatiotemporal Signals

**Marko Mihajlovic**[1], **Sergey Prokudin**[1,3]**, Marc Pollefeys**[1,2]**, Siyu Tang**[1] [*]
ETH Zurich[1]; Microsoft[2]; ROCS, University Hospital Balgrist, University of Zürich [3]

`markomih.github.io/ResFields`

## Abstract

Neural fields, a category of neural networks trained to represent high-frequency signals, have gained significant attention in recent years due to their impressive performance in modeling complex 3D data, such as signed distance (SDFs) or radiance fields (NeRFs), via a single multi-layer perceptron (MLP). However, despite the power and simplicity of representing signals with an MLP, these methods still face challenges when modeling large and complex temporal signals due to the limited capacity of MLPs. In this paper, we propose an effective approach to address this limitation by incorporating temporal residual layers into neural fields, dubbed ResFields. It is a novel class of networks specifically designed to effectively represent complex temporal signals. We conduct a comprehensive analysis of the properties of ResFields and propose a matrix factorization technique to reduce the number of trainable parameters and enhance generalization capabilities. Importantly, our formulation seamlessly integrates with existing MLP-based neural fields and consistently improves results across various challenging tasks: 2D video approximation, dynamic shape modeling via temporal SDFs, and dynamic NeRF reconstruction. Lastly, we demonstrate the practical utility of ResFields by showcasing its effectiveness in capturing dynamic 3D scenes from sparse RGBD cameras of a lightweight capture system.

## 1 Introduction

Multi-layer Perceptron (MLP) is a common neural network architecture used for representing continuous spatiotemporal signals, known as neural fields. Its popularity stems from its capacity to encode continuous signals across arbitrary dimensions (Kim & Adalı, 2003). Additionally, inherent implicit regularization (Goodfellow et al., 2016; Neyshabur et al., 2014) and spectral bias (Rahaman et al., 2019) equip MLPs with excellent interpolation capabilities. Due to these remarkable properties, MLPs have achieved widespread success in many applications such as image synthesis, animation, texture generation, and novel view synthesis (Tewari et al., 2022; Xie et al., 2022).

However, the spectral bias of MLPs (Rahaman et al., 2019), which refers to the tendency of neural networks to learn functions with low frequencies, presents a challenge when it comes to accurately representing complex real-world signals and capturing fine-grained details. Previous efforts have aimed to address the spectral bias by utilizing techniques like positional encoding (Vaswani et al., 2017; Mildenhall et al., 2020; Zhong et al., 2019; Müller et al., 2022) or special activation functions (Sitzmann et al., 2020b; Fathony et al., 2020). However, even with these methods, representing fine-grained details remains a challenge, particularly when dealing with large spatiotemporal signals such as long videos or dynamic 3D scenes.

A straightforward way of increasing the capacity of MLPs is to increase the network complexity in terms of the total number of neurons. However, such an approach would make the inference and optimization slower and more GPU memory expensive, as time and memory complexity scales with respect to the total number of parameters. Another possibility is to meta-learn MLP weights (Sitzmann et al., 2020a) and maintain specialized independent parameters, but this imposes slow training that does not scale to photo-realistic reconstructions (Tancik et al., 2021). By far the most popular approach for increasing modeling capacity is to partition the spatiotemporal field and fit

---

[*]Code, data, and pre-trained models are released at `https://github.com/markomih/ResFields`

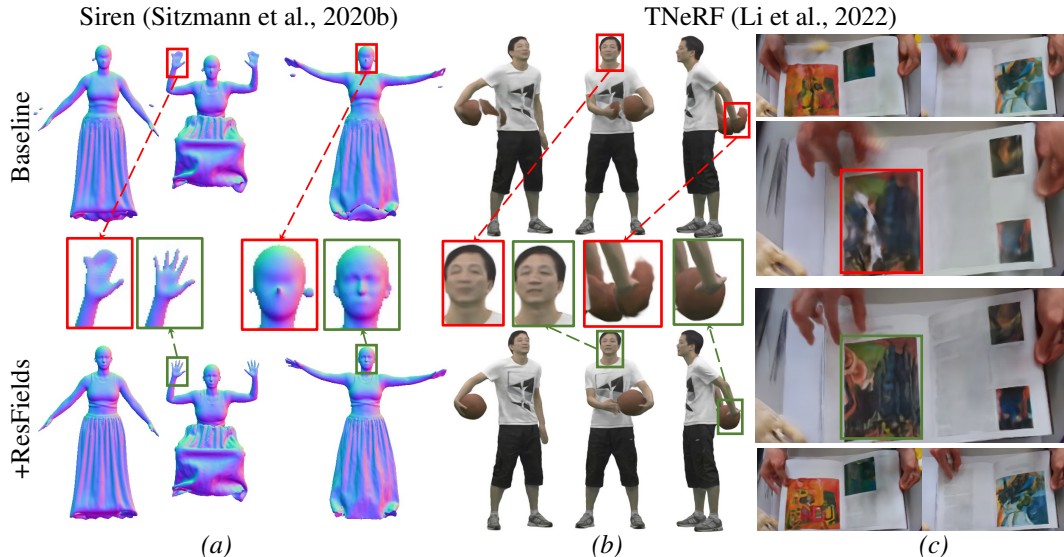

Siren (Sitzmann et al., 2020b)    TNeRF (Li et al., 2022)

*(a)*    *(b)*    *(c)*

Figure 1: **ResField** extends an MLP architecture to effectively represent complex temporal signals by replacing the conventional linear layers with Residual Field Layers. As such, ResField is versatile and straightforwardly compatible with most existing temporal neural fields. Here we demonstrate its applicability on three challenging tasks by extending Siren (Sitzmann et al., 2020b) and TNeRF (Li et al., 2022): *(a)* learning temporal signed distance fields and *(b)* neural radiance fields from four RGB views and *(c)* from three time-synchronized RGBD views captured by our lightweight rig. The figure is best viewed in electronic format on a color screen, please zoom-in to observe details.

separate/local neural fields (Reiser et al., 2021; Müller et al., 2022; Chen et al., 2022). However, these approaches hinder global reasoning and generalization due to local gradient updates of grid structures (Peng et al., 2023).

The challenge that we aim to address is how to increase the model capacity in a way that is agnostic to the design choices of MLP neural fields. This includes architecture, input encoding, and activation functions. At the same time, we must maintain the implicit regularization property of neural networks and retain compatibility with existing techniques developed for reducing the spectral bias (Mildenhall

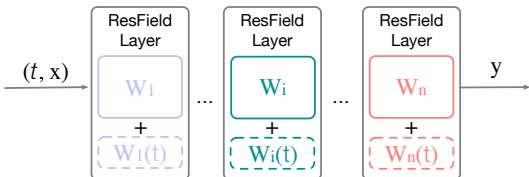

Figure 2: **ResField MLP Architecture.**

et al., 2020; Sitzmann et al., 2020b). Our key idea is to substitute MLP layers with time-dependent layers (see Fig. 2) whose weights are modeled as trainable residual parameters $\mathcal{W}_i(t)$ added to the existing layer weights $\mathbf{W}_i$. We dub neural fields implemented in this way ResFields.

Increasing the model capacity in this way offers three key advantages. First, the underlying MLP does not increase in width and hence, maintains the inference and training speed. This property is crucial for most practical downstream applications of neural fields, including NeRF (Mildenhall et al., 2020) which aims to solve inverse volume rendering (Drebin et al., 1988) by querying neural fields billions of times. Second, this modeling retains the implicit regularization and generalization properties of MLPs, unlike other strategies focused on spatial partitioning (Reiser et al., 2021; Müller et al., 2022; Peng et al., 2023; Işık et al., 2023). Finally, ResFields are versatile, easily extendable, and compatible with most MLP-based methods for spatiotemporal signals.

However, the straightforward implementation of ResFields could lead to reduced interpolation properties due to a large number of unconstrained trainable parameters. To this end, inspired by well-explored low-rank factorized layers (Denil et al., 2013; Ioannou et al., 2015; Khodak et al., 2021), we propose to implement the residual parameters as a global low-rank spanning set and a set of time-dependent coefficients. As we show in the following sections, this modeling enhances the generalization properties and further reduces the memory footprint caused by maintaining additional network parameters.

In summary, our key contributions are:

- We propose an architecture-agnostic building block for modeling spatiotemporal fields that we dub ResFields.
- We systematically demonstrate that our method benefits a number of existing methods: Sitzmann et al. (2020b); Pumarola et al. (2021); Park et al. (2021a;b); Li et al. (2022); Cai et al. (2022); Cao & Johnson (2023); Fridovich-Keil et al. (2023).
- We validate ResFields on four challenging tasks and demonstrate state-of-the-art (Fig. 1): 2D video approximation, temporal 3D shape modeling via signed distance functions, radiance field reconstruction of dynamic scenes from sparse RGB(D) images, and learning 3D scene flow.

## 2   RELATED WORK

**Neural field** is a field – a physical quantity that has a value for every point in time and space – that is parameterized fully or in part by a neural network (Xie et al., 2022), typically an MLP as the universal approximator (Kim & Adalı, 2003). However, straightforward fitting of signals to regular MLPs yields poor reconstruction quality due to the spectral bias of learning low frequencies (Rahaman et al., 2019). Even though this issue has been alleviated through special input encodings (Mildenhall et al., 2020; Barron et al., 2021; 2022) or activation functions (Sitzmann et al., 2020b; Tancik et al., 2020; Fathony et al., 2020; Lindell et al., 2022; Shekarforoush et al., 2022), neural fields still cannot scale to long and complex temporal signals due to the limited capacity. A natural way of increasing the modeling capacity is to increase the network's size in terms of the number of parameters. However, this trivial solution does not scale with GPU and training time requirements.

**Hybrid neural fields** leverage explicit grid-based data structures with learnable feature vectors to improve the modeling capacity via spatial (Takikawa et al., 2021; Müller et al., 2022; Chen et al., 2022; Chan et al., 2022) and temporal (Shao et al., 2023; Fridovich-Keil et al., 2023; Cao & Johnson, 2023; Peng et al., 2023) partitioning techniques. However, these approaches sacrifice the desired global reasoning and implicit regularization (Neyshabur et al., 2014; Goodfellow et al., 2016) that is needed for generalization, especially for solving ill-posed problems like inverse rendering. In contrast, our solution, ResFields, focuses on improving pure neural network-based approaches that still hold state-of-the-art results across several important applications, as we will demonstrate later.

**Input-dependent MLP weights** is another common strategy for increasing the capacity of MLPs by directly regressing MLP weights, e.g. via a hypernetwork (Mehta et al., 2021; Wang et al., 2021c) or a convolutional (Peng et al., 2023) neural network. However, these approaches introduce an additional, much larger network that imposes a significant computational burden for optimizing neural fields. KiloNeRF (Reiser et al., 2021) proposes to speed up the inference of static neural radiance fields by distilling the learned radiance field into a grid of small independent MLPs. However, since a bigger MLP is still used during the first stage of the training, this model has the same scaling limitations as the original NeRF. Closest in spirit to our approach, the level-of-experts (LoE) model (Hao et al., 2022) introduces an input-dependent hierarchical composition of shared MLP weights at the expense of reduced representational capacity. Compared to LoE, ResFields demonstrate stronger generalization and higher representational power for modeling complex spatiotemporal signals.

**Temporal fields** are typically modeled by feeding the time-space coordinate pairs to neural fields. SIREN (Sitzmann et al., 2020b) was one of the first neural methods to faithfully reconstruct a 2D video signal. However, scaling this approach to 4D is infeasible and does not produce desired results as demonstrated in dynamic extensions of NeRF models (Pumarola et al., 2021; Li et al., 2022). Therefore, most of the existing NeRF solutions (Pumarola et al., 2021; Park et al., 2021a) decouple the learning problem into learning a static canonical neural field and a deformation neural network that transforms a query point from the observation to the canonical space where the field is queried. However, these methods tend to fail for more complex signals due to the difficulty of learning complex deformations via a neural network, as observed by Gao et al. (2022). To alleviate the problem, HyperNeRF (Park et al., 2021b) introduced an additional small MLP and per-frame learnable ambient codes to better capture topological variations, increase the modeling capacity, and simplify the learning of complex deformation. The recent NDR (Cai et al., 2022), a follow-up work of HyperNeRF, further improves the deformation field by leveraging invertible neural networks and more constrained SDF-based density formulation (Yariv et al., 2021). All of these methods are fully compatible with the introduced ResFields paradigm which consistently improves baseline results.

**Scene flow** is commonly used to model the dynamics of neural fields. Most works use MLPs to model scene flow by predicting offset vectors (Pumarola et al., 2021; Li et al., 2021b; Prokudin et al., 2023; Wang et al., 2023b), SE(3) transformation (Park et al., 2021a; Wang et al., 2023a), coefficients of pre-defined bases (Wang et al., 2021a; Li et al., 2023), or directly using invertible architectures (Cai et al., 2022; Wang et al., 2023c). These representations are compatible with ResFields which further enhance their learning capabilities.

**Residual connections** have a long history in machine learning. They first appeared in Rosenblatt (1961) in the context of coupled perceptron networks. Rosenblatt's insight was that the residual connections increase the efficiency of responding to input signals. Since then, residual connections have been extensively studied and found a major practical utility as a solution to training deep neural networks by overcoming the vanishing gradient problem (Hochreiter, 1998; Srivastava et al., 2015; He et al., 2016) and became a de facto standard for modeling neural networks. Unlike these residual connections that are added to the output of MLP layers, our ResField layers model the residuals of the MLP weights, which in turn yields higher representation power of neural fields, making them more suitable for modeling complex real-world spatiotemporal signals. To the best of our knowledge, directly optimizing residual or multiplicative correctives of model parameters has been explored in the context of fine-tuning large language models (Karimi Mahabadi et al., 2021; Hu et al., 2021; Dettmers et al., 2023) or predicting model weights (Wang et al., 2021c), and has not been explored for directly training spatiotemporal neural fields.

## 3 ResFields: Residual Neural Fields for Spatiotemporal Signals

**Formulation.** Temporal neural fields encode continuous signals $f : \mathbb{R}^d \times \mathbb{R} \mapsto \mathbb{R}^c$ via a neural network $\Phi_\theta$, where the input is a time-space coordinate pair ($t \in \mathbb{R}$, $\mathbf{x} \in \mathbb{R}^d$) and the output is a field quantity $y \in \mathbb{R}^c$. More formally, the temporal neural field is defined as:

$$\Phi_\theta(t, \mathbf{x}) = \sigma_n\big(\mathbf{W}_n(\phi_{n-1} \circ \phi_{n-2} \circ \cdots \circ \phi_1)(t, \mathbf{x}) + \mathbf{b}_n\big), \tag{1}$$

$$\phi_i(t, \mathbf{x}_i) = \sigma_i(\mathbf{W}_i \mathbf{x}_i + \mathbf{b}_i), \tag{2}$$

where $\phi_i : \mathbb{R}_i^N \mapsto \mathbb{R}_i^M$ is the $i$th layer of the MLP, which consists of the linear transformation by the weight matrix $\mathbf{W}_i \in \mathbb{R}^{N_i \times M_i}$ and the bias $\mathbf{b}_i \in \mathbb{R}^{N_i}$ applied to the input $\mathbf{x}_i \in \mathbb{R}^{M_i}$, followed by a non-linear activation function $\sigma_i$. The network parameters $\theta$ are optimized by minimizing a loss term $\mathcal{L}$ directly w.r.t a ground truth signal or indirectly by relating a field quantity to the sensory input, e.g. via volume rendering equation for radiance field reconstruction.

**Limitations of MLPs.** To model complex and long signals, it is crucial for the underlying MLP to have a sufficient modeling capacity, which scales with the total number of parameters. However, as the MLP size increases, the training time of neural fields becomes slower while increasing the GPU memory requirements, ultimately leading to the bottleneck being the MLP's size. This is especially highlighted for dynamic radiance field reconstruction which requires solving an inverse rendering problem through billions of MLP queries. In the following, we introduce ResFields, an approach for alleviating the capacity bottleneck for modeling and reconstructing spatiotemporal signals.

**ResFields model.** We introduce residual field layers (Fig. 2) to effectively capture large and complex spatiotemporal signals. ResFields, an MLP that uses at least one residual field layer, alleviates the aforementioned capacity bottleneck without increasing the size of MLPs in terms of the number of layers and neurons. In particular, we replace a linear layer of an MLP $\phi_i$ with our temporal time-conditioned residual layer defined as:

$$\phi_i(t, \mathbf{x}_i) = \sigma_i((\mathbf{W}_i + \boldsymbol{\mathcal{W}}_i(t))\mathbf{x}_i + \mathbf{b}_i), \tag{3}$$

where $\boldsymbol{\mathcal{W}}_i(t) : \mathbb{R} \mapsto \mathbb{R}^{N_i \times M_i}$ is time-dependent and models residuals of the network weights. This simple formulation increases the model capacity via additional trainable parameters without modifying the overall network architecture.

**ResFields factorization.** However, naively implementing $\boldsymbol{\mathcal{W}}_i(t) \in \mathbb{R}^{N_i \times M_i}$ as a dictionary of trainable weights would yield a vast amount of independent and unconstrained parameters. This would result in a partitioning of spatiotemporal signal, akin to the

Figure 3: **Factorization of $\boldsymbol{\mathcal{W}}_i$.**

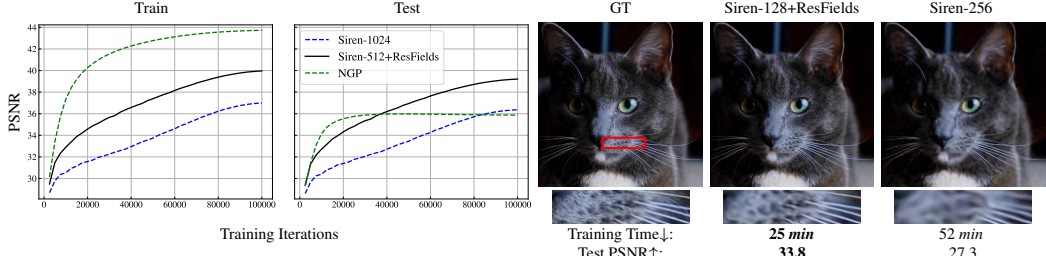

Figure 4: **2D video approximation.** Comparison of different neural fields on fitting RGB videos. The training and test PSNR curves (left and right respectively) indicate the trade-off between the model's capacity and generalization properties. Instant NGP offers good overfitting capabilities, however, it struggles to generalize to unseen pixels. A Siren MLP with 1024 neurons (Siren-1024), shows good generalization properties, however, it lacks representation power (low training and low test PSNR). A smaller Siren with 512 neurons implemented with ResFields (Siren-512+ResFields) demonstrates good generalization while offering higher model capacity. Besides the higher accuracy, our approach offers approximately 2.5 times faster convergence and 30% lower GPU memory requirements due to using a smaller MLP (Tab. 1). Results on the right provide a visual comparison of Siren with 256 neurons and Siren with 128 neurons implemented with ResField layers.

space partitioning methods (Reiser et al., 2021; Müller et al., 2022; Shao et al., 2023), and hinder a global reasoning and implicit bias of MLPs, essential properties for solving under constrained problems such as a novel view synthesis from sparse setups. To this end, inspired by well-established low-rank factorized layers (Denil et al., 2013; Ioannou et al., 2015; Khodak et al., 2021), we directly optimize time-dependent coefficients and $R_i$-dimensional spanning set of residual network weights that are shared across the entire spatiotemporal signal (see Fig. 3). In particular, the residual of network weights are defined as

$$\boldsymbol{\mathcal{W}}_i(t) = \sum\nolimits_{r=1}^{R_i} \mathbf{v}_i(t)[r] \cdot \mathbf{M}_i[r], \tag{4}$$

where the coefficients $\mathbf{v}(t) \in \mathbb{R}^{R_i}$ and the spanning set $\mathbf{M} \in \mathbb{R}^{R_i \times N_i \times M_i}$ are trainable parameters; square brackets denote element selection. To model continuous coefficients over the time dimension, we implement $\mathbf{v} \in \mathbb{R}^{T_i \times R_i}$ as a matrix and linearly interpolate its rows. Such formulation reduces the total number of trainable parameters and further prevents potential undesired overfitting that is common for field partition methods as we will demonstrate later (Sec. 4.5).

**Key idea.** The goal of our parametrization is to achieve high learning capacity while retaining good generalization properties. To achieve this for a limited budget in terms of the number of parameters, we allocate as few independent parameters per time step (in $\mathbf{v}(t)$) and as many globally shared parameters $\mathbf{M}$. Allocating more capacity towards the shared weights will enable *1)* the increased capacity of the model due to its ability to discover small patterns that could be effectively compressed in shared weights and *2)* stronger generalization as the model is aware of the whole sequence through the shared weights. Given these two objectives, we design $\mathbf{v}(t)$) to have very few parameters ($R_i$) and $\mathbf{M}$ to have most parameters ($R_i \times N_i \times M_i$); see the Sup. Mat. for further implementation details.

## 4 EXPERIMENTS

To highlight the versatility of ResFields, we analyze our method on four challenging tasks: 2D video approximation via neural fields, learning of temporal signed distance functions, radiance reconstruction of dynamic scenes from calibrated RGB(D) cameras, and learning 3D scene flow.

### 4.1 2D VIDEO APPROXIMATION

Learning a mapping of pixel coordinates to the corresponding RGB colors is a popular benchmark for evaluating the model capacity of fitting complex signals (Müller et al., 2022; Sitzmann et al., 2020b). For comparison, we use two videos (*bikes* and *cat* from Sitzmann et al. (2020b)) that consist respectively of 250 and 300 frames (with resolutions at $512 \times 512$ and $272 \times 640$) and fit neural representations by minimizing the mean squared error w.r.t ground truth RGB values.

Unlike the proposed setup in Sitzmann et al. (2020b) where the focus is pure overfitting to the image values, our goal is to also evaluate the interpolation aspects of the models. For this, we leave out 10% of randomly sampled pixels for validation and fit the video signal on the remaining ones. We compare our approach against Instant NGP, a popular grid-based approach to neural field modeling, with the best hyperparameter configuration for the task (see supplementary). We also compare against a five-layer Siren network with 1024 neurons (denoted as Siren-1024), as a pure MLP-based approach. For our model, we choose a five-layer Siren network with 512 neurons, whose hidden layers are implemented as residual field layers with the rank $R_i = 10$ for all hidden layers (Siren-512+ResFields). We refer to the supplementary for more details and ablation studies on the number of factors, ranks, and layers for the experiment.

Table 1: **Video approximation.**

|  | test PSNR↑ | $t[it/s]$↑ | GPU↓ |
|---|---|---|---|
| NGP | 34.52 | 131 | 1.6G |
| Siren-1024 | 36.37 | 3.55 | 9.7G |
| Siren-512+ResFields | **39.21** | **9.78** | **6.5G** |

**Insights.** We report training and test PSNR values averaged over the two videos in Fig. 4 and Tab. 1. Here, Instant-NGP offers extremely fast and good overfitting abilities as storing the data in the hash grid structure effectively resolves the problem of limited MLP capacity, alleviating the need for residual weights. This, however, comes at the expense of the decreased generalization capability. Siren-1024 has good generalization properties, but clearly underfits the signal and suffers from blur artifacts. Unlike Siren-1024, Siren-512 with ResFields offers significantly higher reconstruction and generalization quality (36.37 vs 39.21 PSNR) while requiring 30% less GPU memory and being about 2.5 times faster to train.

This simple experiment serves as a proof of concept and highlights our ability to fit complex temporal signals with smaller MLP architectures, which has a significant impact on the practical downstream applications as we discuss in the following sections.

## 4.2 TEMPORAL SIGNED DISTANCE FUNCTIONS (SDF)

Signed-distance functions model the orthogonal distance of a given spatial coordinate $\mathbf{x}$ to the surface of a shape, where the sign indicates whether the point is inside the shape. We model a temporal sequence of signed distance functions via a neural field network that maps a time-space coordinate pair ($t \in \mathbb{R}$, $\mathbf{x} \in \mathbb{R}^3$) to a signed distance value ($y \in \mathbb{R}$).

We sample five sequences of different levels of difficulty (four from Deforming Things (Li et al., 2021a) and one from ReSynth (Ma et al., 2021) and convert the ground-truth meshes to SDF values. We supervise all methods by the MAPE loss following Müller et al. (2022). To benchmark the methods, we extract a sequence of meshes from the learned neural fields via marching cubes (Lorensen & Cline, 1987) and report L1 Chamfer distance (CD↓) and normal consistency (NC↓) w.r.t the ground-truth meshes (scaled by $10^3$ and $10^2$ respectively). As a main baseline, we use the current state-of-the-art Siren network (five layers) and compare it against Siren implemented with our ResField layers,

Table 2: **Temporal SDF.**

|  | Rank $R_i$ | Resources GPU↓ | $t[ms]$↓ | Mean CD↓ | ND↓ |
|---|---|---|---|---|---|
| Siren-128 |  | 2.4G | 20.06 | 15.06 | 27.23 |
| +ResFields | 5 |  |  | 9.47 | 18.54 |
|  | 10 | 2.5G | 20.25 | 8.79 | 16.61 |
|  | 20 |  |  | 8.43 | 15.48 |
|  | 40 |  |  | **8.16** | **14.19** |
| Siren-256 |  | 3.6G | 47.99 | 9.04 | 16.37 |
| +ResFields | 5 |  |  | 7.90 | 13.00 |
|  | 10 | 3.8G | 48.19 | 7.71 | 12.24 |
|  | 20 |  |  | **7.66** | 11.84 |
|  | 40 |  |  | 7.67 | **11.67** |

where residual field layers are applied to three middle layers. We empirically observe that using ResField on the first and last layers has a marginal impact on the performance since weight matrices are small and do not impose a bottleneck for modeling capacity.

**Insights.** Quantitative and qualitative results (Tab. 2, Fig. 1) demonstrate that ResFields consistently improve the reconstruction quality, with the higher rank increasingly improving results. Importantly, we observe that Siren with 128 neurons and ResFields (rank 40), performs better compared to the vanilla Siren with 256 neurons, making our method over two times faster while requiring less GPU memory due to using a much smaller MLP architecture. Alleviating this bottleneck is of utmost importance for the reconstruction tasks that require solving inverse rendering by querying the neural field billions of times as we demonstrate in the next experiment.

## 4.3 TEMPORAL NEURAL RADIANCE FIELDS (NeRF)

Temporal or Dynamic NeRF represents geometry and texture as a neural field that models a function of color and density. The model is trained by minimizing the pixel-wise error metric between the images captured from known camera poses and ones rendered via the differentiable ray marcher

Table 3: **Temporal radiance field** reconstruction on Owlii (Xu et al., 2017). Previous state-of-the-art methods **consistently** benefit from ResField layers without imposing a high computational overhead; bold numbers denote best per-sequence performance while colors denote the overall 1st, 2nd, and 3rd best-performing model; $i$ denotes which layers are substituted with ResFeilds layers.

| | $t\downarrow$ | FPS↑ | Mean CD↓ | SSIM↑ | PSNR↑ | Basketball CD↓ | SSIM↑ | PSNR↑ | Model CD↓ | SSIM↑ | PSNR↑ | Dancer CD↓ | SSIM↑ | PSNR↑ | Exercise CD↓ | SSIM↑ | PSNR↑ |
|---|---|---|---|---|---|---|---|---|---|---|---|---|---|---|---|---|---|
| Neus2 (Wang et al., 2023d) | 0.5h | 30 | 69.4 | 91.01 | 21.73 | 75.7 | 90.57 | 20.48 | 65.6 | 89.64 | 23.01 | 77.1 | 93.16 | 23.23 | 59.2 | 90.67 | 20.21 |
| Tensor4D (Shao et al., 2023) | 15h | 0.035 | 32.8 | 91.05 | 22.59 | 30.5 | 91.22 | 22.51 | 40.3 | 89.30 | 22.46 | 26.7 | 91.53 | 23.24 | 33.5 | 92.16 | 22.16 |
| HexPlane (Cao & Johnson, 2023) | | 0.359 | 20.9 | 92.62 | 24.71 | 17.3 | 93.22 | 25.13 | 24.2 | 91.48 | 25.23 | 23.5 | 91.88 | 23.53 | 18.8 | 93.92 | 24.96 |
| + ResFields ($i=1$) | 5h | 0.357 | 17.8 | 93.51 | 25.61 | 14.9 | 93.96 | 25.91 | **21.3** | 92.58 | **26.19** | 19.7 | 93.16 | 24.86 | 16.3 | 94.30 | 25.33 |
| + ResFields ($i=1,2,3$) | | 0.354 | 17.6 | 93.74 | 25.79 | **14.5** | **94.47** | **26.62** | 21.7 | **92.61** | 25.82 | **18.9** | **93.47** | **25.14** | **15.7** | **94.69** | **25.82** |
| DyNeRF (Li et al., 2022) | | 0.328 | 31.0 | 91.95 | 23.59 | 28.0 | 92.56 | 23.49 | 44.9 | 89.84 | 23.11 | 30.7 | 91.54 | 23.33 | 20.3 | 93.88 | 24.45 |
| + ResFields ($i=1$) | | 0.327 | 20.8 | 93.69 | 25.57 | **14.7** | **94.58** | **26.54** | 26.1 | 92.24 | 25.36 | 24.6 | 93.35 | 25.20 | 17.7 | 94.59 | 25.17 |
| + ResFields ($i=1,2,3$) | 12h | 0.323 | 19.3 | 93.81 | 25.49 | 20.3 | 93.49 | 24.77 | **22.2** | 93.07 | **26.16** | **17.6** | **93.69** | 25.22 | 17.1 | **94.99** | **25.80** |
| + ResFields ($i=1,\dots,7$) | | 0.316 | 19.6 | 94.00 | 25.54 | 17.9 | 94.47 | 25.63 | 23.5 | **93.15** | 26.11 | 20.0 | 93.58 | **25.28** | **16.9** | 94.81 | 25.13 |
| TNeRF (Li et al., 2022) | | 0.339 | 17.2 | 94.18 | 26.18 | 15.1 | 94.57 | 26.33 | 20.2 | 93.31 | 26.52 | 19.3 | 93.53 | 25.09 | 14.1 | 95.33 | 26.77 |
| + ResFields ($i=1$) | | 0.339 | 14.6 | 94.99 | 27.15 | **12.1** | 95.67 | 27.98 | 18.5 | 94.07 | 27.23 | 14.9 | 94.59 | 26.20 | 13.0 | 95.63 | 27.19 |
| + ResFields ($i=1,2,3$) | 12h | 0.334 | 14.2 | 95.21 | 27.44 | 12.2 | 95.84 | **27.98** | 18.3 | 94.33 | **27.81** | **13.3** | 94.87 | 26.55 | 12.9 | 95.82 | 27.40 |
| + ResFields ($i=1,\dots,7$) | | 0.328 | 14.2 | 95.45 | 27.55 | 12.2 | **95.90** | 27.82 | **17.8** | **94.49** | 27.45 | 14.5 | **95.22** | **26.82** | **12.3** | **96.21** | **28.11** |
| DNeRF (Pumarola et al., 2021) | | 0.215 | 32.1 | 92.09 | 23.36 | 22.3 | 93.21 | 24.74 | 44.0 | 90.51 | 23.19 | 38.5 | 91.17 | 21.29 | 23.4 | 93.47 | 24.21 |
| + ResFields ($i=1$) | | 0.214 | 14.2 | 95.16 | 27.33 | 12.1 | 95.88 | 28.26 | 18.1 | 94.15 | 27.03 | 14.1 | 94.66 | 26.24 | 12.8 | 95.95 | 27.79 |
| + ResFields ($i=1,2,3$) | 18h | 0.213 | 14.0 | 95.34 | 27.60 | 12.2 | 95.95 | 28.20 | 17.6 | 94.45 | **27.84** | 14.0 | 94.88 | 26.40 | 12.4 | 96.08 | 27.97 |
| + ResFields ($i=1,\dots,7$) | | 0.210 | 14.0 | 95.67 | 27.89 | **12.0** | **96.15** | **28.34** | 17.3 | **94.85** | 27.83 | 14.3 | **95.45** | 27.25 | **12.3** | **96.21** | **28.14** |
| Nerfies (Park et al., 2021a) | | 0.180 | 23.2 | 93.15 | 24.35 | 21.1 | 93.53 | 24.74 | 28.2 | 92.02 | 24.25 | 23.8 | 92.96 | 23.81 | 19.7 | 94.09 | 24.60 |
| + ResFields ($i=1$) | | 0.180 | 14.6 | 95.12 | 27.26 | 12.3 | 95.64 | 27.86 | 19.3 | 93.95 | 26.91 | 14.2 | 95.10 | 27.00 | 12.7 | 95.77 | 27.29 |
| + ResFields ($i=1,2,3$) | 24h | 0.179 | 14.0 | 95.32 | 27.43 | 11.9 | 95.78 | **27.87** | 18.6 | 94.30 | 27.21 | **13.0** | 95.27 | 27.11 | 12.5 | 95.91 | 27.51 |
| + ResFields ($i=1,\dots,7$) | | 0.177 | 13.8 | 95.57 | 27.72 | **11.8** | **95.79** | 27.42 | 17.6 | **94.68** | **27.78** | 13.5 | **95.67** | **27.73** | **12.2** | **96.16** | **27.94** |
| HyperNeRF (Park et al., 2021b) | | 0.145 | 16.0 | 94.94 | 26.84 | 13.0 | 95.47 | 27.44 | 22.0 | 93.76 | 26.50 | 15.7 | 94.89 | 26.27 | 13.2 | 95.64 | 27.15 |
| + ResFields ($i=1$) | | 0.144 | 14.4 | 95.18 | 27.36 | 12.4 | 95.73 | 28.05 | 18.7 | 94.17 | 27.14 | 14.2 | 95.10 | 26.96 | 13.0 | 95.65 | 27.05 |
| + ResFields ($i=1,2,3$) | 32h | 0.144 | 14.1 | 95.35 | 27.45 | 12.4 | **95.86** | **28.11** | 18.4 | 94.36 | 27.32 | **12.9** | 95.35 | **27.24** | 12.8 | 95.82 | 27.14 |
| + ResFields ($i=1,\dots,7$) | | 0.143 | 14.2 | 95.50 | 27.64 | **11.9** | 95.77 | 27.54 | **18.0** | **94.63** | **27.76** | 14.3 | **95.38** | 27.11 | **12.4** | **96.24** | **28.16** |
| NDR (Cai et al., 2022) | | 0.129 | 15.3 | 94.82 | 26.78 | 13.2 | 95.36 | 27.31 | 19.7 | 93.98 | 26.95 | 15.4 | 94.27 | 25.69 | 12.9 | 95.65 | 27.18 |
| + ResFields ($i=1$) | | 0.129 | 14.7 | 95.14 | 27.16 | 12.6 | 95.74 | 27.87 | 18.2 | 94.17 | 27.28 | 13.4 | 95.15 | 26.96 | 12.9 | 95.84 | 27.17 |
| + ResFields ($i=1,2,3$) | 34h | 0.129 | 14.0 | 95.36 | 27.55 | 12.2 | 96.00 | **28.31** | **18.1** | 94.29 | 27.21 | 13.2 | 95.12 | 26.87 | 12.6 | 96.04 | 27.81 |
| + ResFields ($i=1,\dots,7$) | | 0.127 | 14.2 | 95.56 | 27.81 | **11.9** | **96.02** | 28.02 | 19.3 | **94.48** | **27.51** | **13.1** | **95.38** | **27.11** | **12.4** | **96.36** | **28.61** |

Figure 5: **Temporal radiance fields** on Owlii (Tab. 3); metrics are averaged across all test views.

(Mildenhall et al., 2020). To better model geometry, we adopt the MLP architecture and signed distance field formulation from VolSDF (Yariv et al., 2021) that defines density function as Laplace's cumulative distribution function applied to SDF. We refer readers to the supplementary for the results with the NeRF backbone and further implementation details.

Following (Wang et al., 2021b), all models are supervised by minimizing the pixel-wise difference between the rendered and ground truth colors ($l_1$ error), the rendered opacity and the gt mask (binary cross-entropy), and further adopting the Eikonal (Gropp et al., 2020) mean squared error loss for well-behaved surface reconstruction under the sparse capture setup:

$$\mathcal{L} = \mathcal{L}_{\text{color}} + \lambda_1 \mathcal{L}_{\text{igr}} + \lambda_2 \mathcal{L}_{\text{mask}}. \tag{5}$$

We use four sequences from the Owlii (Xu et al., 2017) dataset to evaluate the methods. Compared to fully synthetic sequences previously utilized for the task (Pumarola et al., 2021), the dynamic Owlii sequences exhibit more rapid and complex high-frequency motions, making it a harder task for MLP-based methods. At the same time, the presence of ground truth 3D scans allows us to evaluate both geometry and appearance reconstruction quality, as compared to the sequences with only RGB data available (Li et al., 2022; Shao et al., 2023). We render 400 RGB training images from four static camera views from 100 frames/time intervals and 100 test images from a rotating camera from 100 frames. We report L1 Chamfer distance (CD↓) (scaled by $10^3$) and the standard image-based metrics (PSNR↑, SSIM↑).

We benchmark recent state-of-the-art methods and their variations implemented with ResField layers of rank ten ($R_i = 10$) – TNeRF (Pumarola et al., 2021; Li et al., 2022), DyNeRF (Li et al., 2022), DNeRF (Pumarola et al., 2021), Nerfies (Park et al., 2021a), HyperNeRF (Park et al., 2021b), NDR (Cai et al., 2022), and HexPlane (Cao & Johnson, 2023; Fridovich-Keil et al., 2023) – as well as a recent timespace-partitioning methods Tensor4D (Shao et al., 2023) and NeuS2 (Wang et al., 2023d) (with default training configurations). Please see the Sup. Mat. for further details.

**Insights**. We report all quantitative and qualitative results in Tab. 3 and Fig. 5. Results demonstrate that our method consistently improves all baseline methods, achieving new state-of-the-art results for sparse multi-view reconstruction of dynamic scenes. We further observe that more ResField layers gradually improve results until the point of saturation ($i = 1, 2, 3$). This experiment confirms that increasing the modeling capacity to a more-than-needed level does not cause overfitting. Importantly, the simplest/cheapest baseline method TNeRF implemented with ResFields performs better than every other more expensive baseline method in the original form. We believe that such speedup and lower memory requirements are of great benefit to the research community, as they enable the use of lower-end hardware for high-fidelity reconstructions. Given this observation, we set up a simple camera rig and captured longer and more complex sequences to better understand the limitations.

**Lightweight capture from three RGBD views.** We capture four sequences (150 frames) via synchronized Azure Kinects (three for reconstruction and one for validation) and compare TNeRF (w. depth supervision), a

Table 4: **Lightweight capture from three RGBD views.**

| | *Mean* | | Book | | Glasses | | Hand | | Writing | |
|---|---|---|---|---|---|---|---|---|---|---|
| | LPIPS↓ | SSIM↑ | LPIPS | SSIM | LPIPS | SSIM | LPIPS | SSIM↑ | LPIPS | SSIM |
| TNeRF | 0.234 | 79.16 | 0.323 | 68.85 | 0.206 | 80.44 | 0.239 | 81.30 | 0.168 | 86.08 |
| +ResFields | **0.203** | **80.00** | **0.284** | **70.84** | **0.164** | **80.65** | **0.210** | **82.09** | **0.155** | **86.43** |

baseline with a good balance between computational complexity and accuracy, and its enhancement with ResFields applied to all middle layers. Quantitative evaluation in terms of mean SSIM↑ and LPIPS↓ (Zhang et al., 2018) reported in Tab. 4 demonstrates that ResFields consistently benefits the reconstruction (see visuals in Fig. 1 and the Sup. video). However, we observe that both methods struggle to capture thin and tiny surfaces such as the cord of sunglasses.

### 4.4 SCENE FLOW

Scene flow models a 3D motion field for every point in space $\mathbf{x}$ and time $t$. We take the same four Deforming Things sequences from Sec 4.2 and learn bi-directional scene flow. We use 80% of tracked mesh vertices to learn the flow and the remaining 20% for evaluation. As a supervision, we use $l_1$ error between the predicted and the ground truth flow vectors. We consider three motion models that predict 1) offset vectors (Prokudin et al., 2023; Li et al., 2021b), 2)

Table 5: **Scene flow.**

| | | it/s ↑ | type | fwd/bwd $l_1$ ↓ |
|---|---|---|---|---|
| 128 neurons | ReLU MLP | | offset | 6.88 / 7.31 |
| | +ResFields | | | **3.85 / 3.85** |
| | ReLU MLP | 16.5 | SE(3) | 4.57 / 4.58 |
| | +ResFields | | | **2.64 / 2.56** |
| | ReLU MLP | | DCT | 3.19 / 3.19 |
| | +ResFields | | | **2.18 / 2.19** |
| 256 neurons | ReLU MLP | | offset | 6.50 / 6.44 |
| | +ResFields | | | **4.43 / 4.59** |
| | ReLU MLP | 5.6 | SE(3) | 4.00 / 4.10 |
| | +ResFields | | | **2.88 / 2.84** |
| | ReLU MLP | | DCT | 2.47 / 2.48 |
| | +ResFields | | | **1.79 / 1.79** |

SE(3) transformation (Park et al., 2021a; Wang et al., 2023a), and 3) coefficients of the DCT bases (Wang et al., 2021a; Li et al., 2023). For all of them, we consider the same architecture from (Wang et al., 2021a): 8-layer ReLU-MLP with positional encoding.

**Insight.** Tab. 5 reports the $l_1$ error (scaled by $10^3$) for the evaluation points. ResFields greatly benefits learning scene flow across all settings. Moreover, the architecture of 256 neurons is less powerful than its much smaller and faster counterpart (128 neurons) with ResFields (rank of 10) for both forward and backward flow while being around **three times** faster: 16.5 *vs.* 5.6 train it/s.

## 4.5 ABLATION STUDY

**ResField modeling (Tab. 6).** Residual connections on the layer weights $(\mathbf{W}_i + \boldsymbol{\mathcal{W}}_i(t))$ are more powerful compared to modeling residuals on the layer output that is commonly used for conditional generation (Karras et al., 2020), directly modulating layer weights $(\mathbf{W}_i \odot \boldsymbol{\mathcal{W}}_i(t))$ (Mehta et al., 2021), or using time-dependent weights $(\boldsymbol{\mathcal{W}}_i(t))$ as in LoE (Hao et al., 2022). Tab. 6 summarizes the results of these variations on the video approximation task from Sec. 4.1.

**Factorization techniques (Tab. 7).** We compare our factorization (Eq. 4) with alternative techniques: no factorization (Reiser et al., 2021), low-rank matrix-matrix decomposition $(\mathbf{v}(t) \in \mathbb{R}^{N_i \times R_i}, \mathbf{M} \in \mathbb{R}^{R_i \times M_i})$, regressing network parameters (Ha et al., 2017), hierarchical Levels-of-Experts (LoE) (Hao et al., 2022), and the classic CP (Carroll & Chang, 1970) and Tucker (1966). CP and Tucker with varying ranks demonstrate good generalization and overfitting results. No factorization achieves great training PSNR, but its generalization performance is suboptimal which has been mitigated by the hierarchical formulation of LoE. The proposed factorization achieves the best generalization properties. The reported numbers in Tab. 7 are measured on the video approximation task for 30% of unseen pixels. See the Sup. Mat. for additional comparisons.

**Limitations.** Overall ResFields benefits spatiotemporal neural fields when the bottleneck lies in the modeling capacity rather than in solving unconstrained problems. Specifically, we do not observe an advantage on challenging ill-posed monocular reconstruction (Gao et al., 2022) when the main bottleneck is the lack of constraints rather than the network's capacity.

Table 6: **ResField modeling.**

| | *Mean* PSNR↑ | |
| --- | --- | --- |
| | test | train |
| Siren-512 | | |
| $\phi_i(t, \mathbf{x}_i) = \sigma_i(\mathbf{W}_i\mathbf{x}_i + \mathbf{b}_i)$ | 31.89 | 32.13 |
| +output residual weights (Karras et al., 2020) | | |
| $\phi_i(t, \mathbf{x}_i) = \sigma_i(\mathbf{W}_i\mathbf{x}_i + \mathbf{b}_i) + \boldsymbol{\mathcal{W}}_i(t)$ | 32.84 | 33.12 |
| +modulated weights (Mehta et al., 2021) | | |
| $\phi_i(t, \mathbf{x}_i) = \sigma_i((\mathbf{W}_i \odot \boldsymbol{\mathcal{W}}_i(t))\mathbf{x}_i + \mathbf{b}_i)$ | 32.65 | 32.90 |
| +direct (Hao et al., 2022) $\boldsymbol{\mathcal{W}}_i(t)$ | | |
| $\phi_i(t, \mathbf{x}_i) = \sigma_i(\boldsymbol{\mathcal{W}}_i(t)\mathbf{x}_i + \mathbf{b}_i)$ | 35.17 | 35.95 |
| +ResFields | | |
| $\phi_i(t, \mathbf{x}_i) = \sigma_i((\mathbf{W}_i + \boldsymbol{\mathcal{W}}_i(t))\mathbf{x}_i + \mathbf{b}_i)$ | **39.21** | **39.97** |

Table 7: **Factorization techniques.**

| | Factorization | Rank | #params [M] | *Mean* PSNR↑ test | train |
| --- | --- | --- | --- | --- | --- |
| | Siren | | 0.8 | 31.96 | 32.29 |
| | None Reiser et al. (2021) | | 236 | 38.52 | 48.46 |
| | Low-rank matrix-matrix: $\mathbf{v}(t) \in \mathbb{R}^{N_i \times R_i}$ $\mathbf{M} \in \mathbb{R}^{R_i \times M_i}$ | 10 | 5.4 | 35.22 | 36.35 |
| | | 20 | 10.0 | 35.88 | 37.50 |
| | | 40 | 19.3 | 36.67 | 39.01 |
| | | 80 | 37.8 | 37.69 | 41.07 |
| | HyperNetwork Ha et al. (2017) | | 10.6 | 38.60 | 39.56 |
| +ResFields | CP Carroll & Chang (1970) | 10 | 0.8 | 33.04 | 33.36 |
| | | 20 | 0.9 | 33.14 | 33.47 |
| | | 40 | 1.0 | 33.41 | 33.75 |
| | | 80 | 1.1 | 33.72 | 34.08 |
| | Tucker (1966) | 10,64,64 | 1.1 | 33.96 | 34.31 |
| | | 40,64,64 | 1.5 | 34.67 | 35.10 |
| | | 80,64,64 | 2.0 | 35.08 | 35.59 |
| | | 10,256,256 | 3.6 | 36.31 | 36.90 |
| | | 40,256,256 | 9.5 | 38.31 | 39.33 |
| | | 80,256,256 | 17.4 | 39.04 | 40.39 |
| | LoE Hao et al. (2022) | (2,4,8) | 4.5 | 36.42 | 37.37 |
| | | (8,16,32) | 15.5 | 39.87 | 42.27 |
| | | (16,32,64) | 30.2 | 40.53 | 44.15 |
| | | (32,64,128) | 59.5 | 40.62 | 46.35 |
| | Ours Eq. 3 | 10 | 8.7 | 39.59 | 40.80 |
| | | 20 | 16.5 | 40.87 | 42.45 |
| | | 40 | 32.3 | 41.69 | 43.72 |
| | | 80 | 63.8 | 41.51 | 44.39 |

## 5 DISCUSSION AND CONCLUSION

We present a novel approach to overcome the limitations of spatiotemporal neural fields in effectively modeling long and complex temporal signals. Our key idea is to incorporate temporal residual layers into neural fields, dubbed ResFields. The advantage and utility of the method lie in its versatility and straightforward integration into existing works for modeling 2D and 3D temporal fields. ResFields increase the capacity of MLPs without expanding the network architecture in terms of the number of layers and neurons, which allows us to use smaller MLPs without sacrificing the reconstruction quality while achieving faster inference and training time with a lower GPU memory requirement. We believe that progress towards using lower-cost hardware is the key to democratizing research and making technology more accessible. We hope that our study contributes to development of neural fields and provides valuable insights for modeling signals. This, in turn, can lead to advancements in various domains, including computer graphics, computer vision, and robotics.

**Acknowledgments and Disclosure of Funding.** We thank Hongrui Cai and Ruizhi Shao for providing additional details about the baseline methods and Anpei Chen, Shaofei Wang, Songyou Peng, and Theodora Kontogianni for constructive feedback and proofreading the manuscript. This project has been supported by the Innosuisse Flagship project PROFICIENCY No. PFFS-21-19.

**Ethics Statement.** In our pursuit of advancing signal modeling and representation techniques, our work holds the potential to bring positive advancements to various domains within the entertainment and AI industries, benefiting both research and practical applications. However, it is crucial to acknowledge the indirect influence of our efforts on the field of deep fakes, as our methodology contributes to the enhancement of photorealistic reconstruction from images.

**Reproducibility.** In our commitment to promoting openness and transparency in research, we provide comprehensive resources for replicating our results: *1)* Open source code: we will release the source code used in our experiments, which can be found in the supplementary material. This code includes detailed documentation and instructions to facilitate the replication of our main results. *2)* Pre-trained models: we will release our trained models to improve verifiability. *3)* Dataset: our captured dataset will be made publicly available. *4)* Supplementary documentation: in addition to the code, we provide a supplementary document that offers a deeper insight into our experimental setups, training techniques, and other crucial details.

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

# A APPENDIX

We provide additional implementation details and experiments to complement our study. All the reported runtime in this paper is measured on an NVIDIA RTX 3090 GPU card.

## A.1 IMPLEMENTATION DETAILS

**Initialization.** For experiments that use Siren networks (Sitzmann et al., 2020b) (sections 4.1 and 4.2), we follow their proposed initialization scheme. Models used for SDF-based dynamic radiance field reconstruction (Sec. 4.3) are initialized following the geometric initialization scheme (Gropp et al., 2020). Other neural network weights are initialized following Glorot & Bengio (2010).

**Residual weights.** Parameters $(\mathbf{v}_i, \mathbf{M}_i)$ which model our residual weights are initialized with a normal distribution $\sim \mathcal{N}(0, 10^{-2})$ to ensure a negligible modification of the initial MLP weights. We observe that larger initial values may negatively affect geometric and Siren initialization. For all experiments in the main paper, we set the number of coefficients $T_i$ to the number of frames unless specified otherwise.

**Training details.** All models are trained with the Adam optimizer (Kingma & Ba, 2015) with default parameters defined by the PyTorch framework (Paszke et al., 2019). We observe stable convergence with the learning rate of $5 \times 10^{-4}$ and gradual cosine annealing (Loshchilov & Hutter, 2016) until the minimum learning rate of $5 \times 10^{-5}$ for the experiments on dynamic neural radiance fields (Sec. 4.3). For other experiments (sections 4.1 and 4.2), we use the learning rate of $5 \times 10^{-5}$ and cosine annealing until $5 \times 10^{-6}$. All methods are trained respectively for $10^5$, $2 \times 10^5$, $4 \times 10^5$, and $6 \times 10^5$ iterations on the 2D video approximation task (Sec. 4.1), temporal SDF reconstruction (Sec. 4.2), and dynamic volumetric reconstruction (Sec. 4.3) on Owlii (Xu et al., 2017) and our captured sequences. An exception in Sec. 4.3 is with the grid-partitioning methods – Tenso4D (Shao et al., 2023) and HexPlane (Cao & Johnson, 2023; Fridovich-Keil et al., 2023) – which were trained for fewer iterations ($2 \times 10^5$) as they use shallower MLPs which converge faster.

## A.2 2D VIDEO APPROXIMATION TASK

All methods presented in the paper (Sec. 4.1) on the 2D video approximation task are trained for 100k iterations, each iteration containing 200k random samples from the training set.

For the NGP baseline in Sec. 4.1, we follow the default setup and use a two-layer fully fused network with ReLU activation functions and run a grid search of hyperparameters to find the optimal configuration. Specifically, we vary the table size $T$ and the number of levels $L$ in Tab. A.5 and found that the best results are achieved with $T = 23$ and $L = 8$ which is reported in the main paper (Fig. 4). Furthermore, we provide results (Tab. A.5) of a much larger five-layer Siren network with 1700 neurons per layer to match the number of trainable parameters to ResFields implemented with a five-layer Siren network with 512 neurons, each containing 8.7M parameters. Expectedly, we observe that both methods achieve similar fitting and generalization performance. However, training such a huge MLP with 1700 neurons becomes impractical, making our approach over six times faster to train while requiring over two times less GPU memory.

**Number of factors (Tab. A.1).** We further ablate the impact of the number of factors $T_i$ of ResFields. In this experiment, we leave out 10% of randomly sampled pixels for validation and vary the number of factors used for parameterizing the coefficients $\mathbf{v} \in \mathbb{R}^{T_i \times R_i}$, in particular, we set $T_i$ as the percentage of the total number of frames. The results averaged over two videos are reported in Tab. A.1. We observe that the best performance is achieved for 95% when there's little overlap between the co-

Table A.1: **Number of factors.**

| | Factors $T_i$ | *Mean* PSNR↑ test | train |
|---|---|---|---|
| Siren-512 | | 32.02 | 32.27 |
| +ResFields | 100% | 39.86 | 40.73 |
| | 95% | **39.90** | 40.77 |
| | 90% | 39.79 | 40.69 |
| | 80% | 39.69 | 40.62 |
| | 70% | 39.60 | 40.49 |
| | 60% | 39.53 | 40.44 |
| | 50% | 39.45 | 40.37 |
| | 40% | 39.25 | 40.20 |
| | 30% | 39.10 | 40.04 |
| | 20% | 38.87 | 39.82 |
| | 10% | 38.34 | 39.29 |

Table A.2: **Time interpolation.**

| | Factors $T_i$ | *Mean* PSNR↑ test | train |
|---|---|---|---|
| Siren-512 | | 26.72 | 32.36 |
| +ResFields | 90 % | 21.61 | 40.90 |
| | 80 % | 22.01 | 40.82 |
| | 70 % | 24.57 | 40.76 |
| | 60 % | 26.06 | 40.62 |
| | 50 % | 26.12 | 40.58 |
| | 40 % | 25.54 | 40.41 |
| | 30 % | 26.51 | 40.18 |
| | 20 % | 27.32 | 39.91 |
| | 10 % | **27.34** | 39.37 |

Table A.3: **Layers *vs.* rank.**

| ResField Layers $i$ | Rank $R_i$ | #params [M] | *Mean* PSNR↑ test | train |
|---|---|---|---|---|
| 2 | 15 | 4.7 | 37.53 | 38.25 |
| 1, 2, 3 | 5 | | **38.01** | **38.67** |
| 2 | 30 | 8.7 | 38.75 | 39.69 |
| 1, 2, 3 | 10 | | **39.86** | **40.73** |
| 2 | 45 | 12.6 | 39.33 | 40.43 |
| 1, 2, 3 | 15 | | **40.62** | **41.66** |
| 2 | 60 | 16.5 | 39.67 | 40.88 |
| 1, 2, 3 | 20 | | **41.20** | **42.34** |

Table A.4: **Ablation study of different fractions of unseen pixels** on the video approximation task. ResFields consistently demonstrate good generalization properties regardless of the difficulty level.

| | Unseen pixels | *Mean* PSNR↑ test | train | Cat PSNR↑ test | train | Bicycles PSNR↑ test | train |
|---|---|---|---|---|---|---|---|
| Siren-512 | | 32.02 | 32.27 | 31.21 | 31.41 | 32.84 | 33.13 |
| + ResFields | 10% | 39.86 | 40.73 | 38.58 | 39.15 | 41.13 | 42.32 |
| Siren-1024 | | 36.67 | 37.36 | 34.95 | 35.52 | 38.38 | 39.19 |
| + ResFields | | **43.15** | 44.75 | 42.49 | 43.53 | 43.82 | 45.98 |
| Siren-512 | | 31.99 | 32.27 | 31.18 | 31.41 | 32.79 | 33.13 |
| + ResFields | 20% | 39.74 | 40.75 | 38.50 | 39.15 | 40.98 | 42.35 |
| Siren-1024 | | 36.60 | 37.39 | 34.90 | 35.55 | 38.30 | 39.23 |
| + ResFields | | **42.95** | 44.82 | 42.30 | 43.53 | 43.59 | 46.12 |
| Siren-512 | | 31.97 | 32.3 | 31.15 | 31.42 | 32.80 | 33.18 |
| + ResFields | 30% | 39.59 | 40.8 | 38.39 | 39.17 | 40.80 | 42.43 |
| Siren-1024 | | 36.51 | 37.42 | 34.83 | 35.58 | 38.19 | 39.27 |
| + ResFields | | **42.72** | 44.96 | 42.16 | 43.60 | 43.28 | 46.32 |
| Siren-512 | | 31.91 | 32.29 | 31.10 | 31.41 | 32.71 | 33.17 |
| + ResFields | 40% | 39.39 | 40.85 | 38.28 | 39.20 | 40.51 | 42.50 |
| Siren-1024 | | 36.41 | 37.50 | 34.74 | 35.63 | 38.08 | 39.38 |
| + ResFields | | **42.33** | 45.14 | 41.86 | 43.67 | 42.79 | 46.62 |
| Siren-512 | | 31.85 | 32.31 | 31.05 | 31.42 | 32.66 | 33.20 |
| + ResFields | 50% | 39.09 | 40.95 | 38.08 | 39.24 | 40.10 | 42.66 |
| Siren-1024 | | 36.26 | 37.61 | 34.62 | 35.71 | 37.90 | 39.51 |
| + ResFields | | **41.75** | 45.41 | 41.43 | 43.79 | 42.08 | 47.03 |
| Siren-512 | | 31.59 | 32.40 | 30.83 | 31.48 | 32.35 | 33.32 |
| + ResFields | 70% | 37.70 | 41.49 | 37.14 | 39.42 | 38.26 | 43.55 |
| Siren-1024 | | 35.54 | 38.04 | 34.08 | 36.06 | 36.99 | 40.02 |
| + ResFields | | **38.96** | 46.61 | 39.00 | 44.44 | 38.91 | 48.78 |

efficients. In practice, there is a negligible difference compared to using independent coefficients (100%) which we use as a default configuration as it is slightly computationally faster.

**Time interpolation (Tab. A.2)**. One downside of using per-frame independent coefficients is that it does not support time interpolation. We conduct an experiment to evaluate the interpolation along the time axis. Here we randomly sample 10% of frames and leave them out for validation. As expected, the lower number of factors $T_i$ leads to a greater overlap among the frames, consequently leading to better interpolation properties results, while gradually decreasing the training PSNR.

**Layers *vs.* rank (Tab. A.3)**. Another natural question to consider is whether it is more beneficial to have more ResField layers or a single ResFied layer with a higher rank while maintaining the constant number of trainable parameters. We conduct this experiment on the video approximation task and compare methods with an equal number of parameters. We conclude that multiple ResField layers provide greater modeling capacity.

**Ablation for fewer training samples (Tab. A.4)**. To complete our study and better understand the implicit bias of our method, we further benchmark Siren and ResFields with varying difficulty levels, ranging from 10-70% of unseen pixels. We observe that ResFields consistently demonstrate good generalization across all the levels of difficulty, well above the baseline.

## A.3 TEMPORAL SIGNED DISTANCE FUNCTIONS (SDF)

The architecture of the Siren MLP used for this experiment is identical to the one used for the video approximation task. All methods are trained for 200k iterations, each batch containing 200k samples uniformly sampled across time. For each frame we follow the sampling strategy for static SDFs (Müller et al., 2022) and sample 50% of points on the mesh, 37.5% normally distributed around the surface $\mathcal{N}(0, 10^{-2})$, and remaining 12.5% are randomly sampled in space.

We provide the full breakdown of per-sequence results in Tab. A.6.

## A.4 TEMPORAL NEURAL RADIANCE FIELDS (NERF)

All methods on the Owlii dataset are trained for 400k iterations, except HexPlanes (Cao & Johnson, 2023; Fridovich-Keil et al., 2023) which converges faster (400k iterations) due to using shallower MLPs, and Tensor4D and NeuS2 for which we follow the default training scheme as the methods require a particular training strategy. The main baselines (TNeRF, DyNeRF, DNeRF, Nerfies, Hy-

Table A.5: **Extended ablation study on the video approximation task.** Siren+ResFields with 512 neurons and Siren with 1700 neurons have an equal number of parameters, however, optimizing a huge MLP with 1700 neurons comes with a great computational cost. Our method achieves **over six times faster training** while requiring **over two times less GPU memory**.

| | | Resources | | | Mean | | Cat Video | | Bikes Video | |
|---|---|---|---|---|---|---|---|---|---|---|
| | | $t$ [it/s] | GPU [G] | #params [M] | test PSNR↑ | train PSNR↑ | test PSNR↑ | train PSNR↑ | test PSNR↑ | train PSNR↑ |
| Siren-512 | | 11.66 | 5.1 | 0.8 | 31.89 | 32.13 | 31.09 | 31.29 | 32.68 | 32.98 |
| +ResFields | R=10 | 9.78 | 6.5 | 8.7 | **39.21** | 39.97 | 37.96 | 38.44 | 40.46 | 41.50 |
| Siren-1700 | | 1.42 | 15 | 8.7 | 39.15 | 40.20 | 37.26 | 38.14 | 41.04 | 42.25 |
| NGP T=20 | L=6 | 150 | 1.3 | 3.6 | 31.27 | 33.25 | 30.23 | 31.71 | 32.31 | 34.78 |
| | L=7 | 153 | 1.3 | 5.7 | 32.08 | 35.06 | 30.89 | 33.42 | 33.28 | 36.70 |
| | L=8 | 157 | 1.2 | 7.8 | 32.61 | 36.64 | 31.34 | 35.15 | 33.88 | 38.13 |
| | L=9 | 158 | 1.1 | 9.9 | 32.41 | 37.80 | 31.06 | 36.21 | 33.75 | 39.39 |
| NGP T=21 | L=6 | 126 | 1.6 | 5.1 | 32.02 | 34.48 | 30.51 | 32.20 | 33.52 | 36.76 |
| | L=7 | 141 | 1.5 | 9.3 | 32.91 | 36.87 | 31.45 | 34.69 | 34.37 | 39.04 |
| | L=8 | 146 | 1.3 | 13.5 | 33.66 | 39.33 | 31.97 | 37.29 | 35.34 | 41.37 |
| | L=9 | 157 | 1.2 | 17.7 | 33.53 | 41.36 | 31.89 | 39.38 | 35.17 | 43.34 |
| NGP T=22 | L=6 | 101 | 2.1 | 5.1 | 32.01 | 34.47 | 30.51 | 32.20 | 33.51 | 36.73 |
| | L=7 | 116 | 1.7 | 13.5 | 33.39 | 37.82 | 31.82 | 35.39 | 34.96 | 40.25 |
| | L=8 | 144 | 1.5 | 21.9 | 34.02 | 41.09 | 32.25 | 38.71 | 35.79 | 43.48 |
| | L=9 | 156 | 1.2 | 30.3 | 33.73 | 43.83 | 32.17 | 41.69 | 35.29 | 45.96 |
| NGP T=23 | L=6 | 75 | 2.7 | 5.1 | 32.02 | 34.48 | 30.52 | 32.20 | 33.52 | 36.76 |
| | L=7 | 99 | 2.1 | 17.4 | 33.83 | 39.51 | 32.14 | 36.44 | 35.52 | 42.57 |
| | L=8 | 131 | 1.6 | 34.2 | 34.52 | 43.85 | 32.91 | 40.85 | 36.13 | 46.85 |
| | L=9 | 148 | 1.2 | 51 | 33.96 | 47.56 | 32.64 | 44.63 | 35.28 | 50.50 |
| NGP T=24 | L=6 | 51 | 3.8 | 5.1 | 32.01 | 34.46 | 30.51 | 32.20 | 33.51 | 36.72 |
| | L=7 | 77 | 2.7 | 17.4 | 33.84 | 39.51 | 32.14 | 36.45 | 35.54 | 42.58 |
| | L=8 | 130 | 1.6 | 51.0 | 34.27 | 44.68 | 32.72 | 41.71 | 35.81 | 47.66 |
| | L=9 | 145 | 1.2 | 84.5 | 34.02 | 49.51 | 33.08 | 46.58 | 34.95 | 52.43 |

Table A.6: **Temporal signed distance function.** Siren implemented with our Residual Field layers consistently improves the reconstruction quality. Moreover Siren with 128 neurons and ResFields (rank 40) performs better compared to the much bigger vanilla Siren with 256 neurons. Hence leading to over two times faster inference and convergence time while maintaining lower GPU memory requirements; $t[ms]$ denotes the average inference time for one million query points. Note that using higher ranks almost does not affect the overall time complexity as the main bottleneck is the total number of queries and neurons.

| | Resources | | Mean | | Bear | | Tiger | | Vampire | | Vanguard | | ReSynth | |
|---|---|---|---|---|---|---|---|---|---|---|---|---|---|---|
| | GPU↓ | $t$[ms]↓ | CD↓ | ND↓ | CD↓ | ND↓ | CD↓ | ND↓ | CD↓ | ND↓ | CD↓ | ND↓ | CD↓ | ND↓ |
| Siren-128 | 2.4G | 20.06 | 15.063 | 27.23 | 7.605 | 4.519 | 5.159 | 4.934 | 29.490 | 63.686 | 17.099 | 42.953 | 15.960 | 20.057 |
| + ResFields (rank=05) | | | 9.471 | 18.537 | 6.813 | 3.569 | 4.422 | 3.639 | 12.105 | 39.457 | 11.420 | 30.414 | 12.594 | 15.606 |
| + ResFields (rank=10) | 2.5G | 20.25 | 8.785 | 16.608 | 6.671 | 3.350 | 4.351 | 3.435 | 9.545 | 32.220 | 11.140 | 29.961 | 12.216 | 14.075 |
| + ResFields (rank=20) | | | 8.427 | 15.483 | 6.659 | 3.301 | 4.325 | 3.328 | 8.708 | 29.661 | 10.465 | 27.541 | 11.980 | 13.584 |
| + ResFields (rank=40) | | | 8.158 | 14.195 | 6.579 | 3.201 | 4.278 | 3.148 | 8.563 | 29.064 | 9.729 | 23.564 | 11.640 | 11.999 |
| Siren-256 | 3.6G | 47.99 | 9.040 | 16.373 | 6.532 | 3.115 | 4.241 | 3.055 | 11.800 | 36.666 | 10.623 | 26.407 | 12.004 | 12.622 |
| + ResFields (rank=05) | | | 7.901 | 13.000 | 6.430 | 3.009 | 4.177 | 2.854 | 8.576 | 28.846 | 8.993 | 20.168 | 11.331 | 10.124 |
| + ResFields (rank=10) | 3.8G | 48.19 | 7.714 | 12.242 | 6.408 | 3.001 | 4.161 | 2.814 | 8.136 | 27.404 | 8.757 | 18.950 | 11.110 | 9.040 |
| + ResFields (rank=20) | | | 7.662 | 11.840 | 6.396 | 2.995 | 4.141 | 2.753 | 8.076 | 27.013 | 8.650 | 18.056 | 11.050 | 8.385 |
| + ResFields (rank=40) | | | 7.675 | 11.674 | 6.381 | 3.070 | 4.137 | 2.723 | 8.243 | 27.296 | 8.525 | 17.526 | 11.087 | 7.755 |

perNeRF, and NDR) are implemented with the MLP architecture from VolSDF (Yariv et al., 2021), however, we reduced the original MLP size from 256 to 128 neurons as it is impractical to train such large MLPs on more expensive multi-view temporal sequences. The SDF MLP is followed by a two-layer color MLP that takes the output feature of the SDF network. Different from the original color MLP that is conditioned on the viewing ray direction, we do not pass this information to the network as it is impossible to capture any view-dependent appearance for this extremely sparse setup. We observe that training without the viewing direction stabilizes training of the baselines, especially those that rely on a deformation network. We follow the original formulation of the flow MLPs used in DNeRF, Nerfies, HyperNeRF, and NDR.

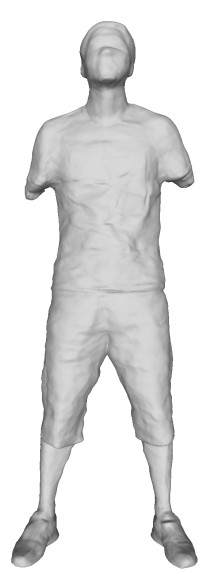

Note that the original formulation of the HexPlane method (Cao & Johnson, 2023; Fridovich-Keil et al., 2023) is not suited for reconstruction from sparse views mainly due to the lack of spherical initialization. Hence we adopt the geometric initialization developed for Triplanes proposed in PET-NeuS (Wang et al., 2023e). Unlike the other MLP-based methods, the SDF MLP head of HexPlane has four linear layers with ReLU activation functions. Furthermore, to reduce the grid-like artifacts caused by space-time discretization, we tune the resolution of planes (128 grid locations per dimension) and employ the additional total variation loss akin to (Cao & Johnson, 2023). We observe that using a higher resolution for six planes impairs the reconstruction quality.

To make the comparison fair among the baselines, for rendering we employ a non-biased uniform sampling along the ray and sample 1100 rays during the training. On each ray, we sample 256 points where the starting and exiting points of the ray are calculated by ray-box intersection. The box for each sequence is estimated from the ground truth scans with a small padding of 5%. All the methods are supervised by minimizing the loss term in Eq. 5, where we set $\lambda_1$ and $\lambda_2$ to 0.1.

In practice, the SDF-based density formulation performs better under a sparse setup due to well-behaved surfaces. However, for completeness, we repeat this experiment with the original NeRF formulation (Tab. A.7). The results demonstrate that all of the baselines consistently benefit from ResFields, making Nerfies+ResFields the overall best-performing method in terms of geometry and HyperNeRF+ResFields in terms of appearance.

Figure A.1: **Base MLP weights**.

**Interpreting residual weights.** To better understand the internal workings of ResField MLPs, we extract a mesh from the base MLP weights without the residual parameters $\mathcal{W}_i(t)$ – see Fig. A.1 for results on the basketball sequence. The extracted mesh demonstrates that the base MLP successfully discovers a pattern that is shared among all frames.

## A.5 Practical reconstruction from a lightweight capture system

We follow the VolSDF architecture from the experiment on the Owlii dataset and train all methods for $600k$ iterations (2200 rays per batch) since the sequences are longer and images are of higher resolution ($540 \times 960$).

As the scenes are forward-facing and not constrained from the opposite side, we employ the depth (Cai et al., 2022) and the sparseness loss $\mathcal{L}_{\text{sparse}}$ (Long et al., 2022). In sum, the final loss term is:

$$\mathcal{L} = \mathcal{L}_{\text{color}} + \lambda_1 \mathcal{L}_{\text{depth}} + \lambda_2 \mathcal{L}_{\text{igr}} + \lambda_3 \mathcal{L}_{\text{sparse}} , \quad \text{(A.1)}$$

where we set $\lambda_1$, $\lambda_2$, and $\lambda_3$ to 0.1 and activate $\mathcal{L}_{\text{sparse}}$ after $70k$ iterations.

Figure A.2: **Data capture**.

**Data capture.** Inspired by the lightweight practical capture of surgeries[1] we create an identical camera rig with four cameras (see Fig. A.2), three for training and one for evaluation. Additionally, we crop captured images by projecting an approximated 3D bounding box around the dynamic region as the main focus of our approach is reconstructing dynamic sequences.

---

[1]OR-X Setup

Table A.7: **Temporal radiance field reconstruction** on the Owlii dataset (Xu et al., 2017) with the NeRF parametrization. Previous state-of-the-art methods consistently benefit from our residual field layers without the computational overhead; results with the VolSDF parametrization are provided in Tab. 3. Colors denote the overall 1st , 2nd , and 3rd best-performing model; $i$ denotes which layers are substituted with ResFeilds layers ($R_i = 10$).

| | Mean CD↓ | SSIM↑ | PSNR↑ | Basketball CD↓ | SSIM↑ | PSNR↑ | Model CD↓ | SSIM↑ | PSNR↑ | Dancer CD↓ | SSIM↑ | PSNR↑ | Exercise CD↓ | SSIM↑ | PSNR↑ |
|---|---|---|---|---|---|---|---|---|---|---|---|---|---|---|---|
| TNeRF (Li et al., 2022) | 61.6 | 93.90 | 26.04 | 70.9 | 94.06 | 25.88 | 52.9 | 93.18 | 27.06 | 65.4 | 93.49 | 25.22 | 57.1 | 94.88 | 25.99 |
| + ResFields ($i=1$) | 53.6 | 94.54 | 26.72 | 53.8 | 95.08 | 27.15 | 53.4 | 93.55 | 27.32 | 53.3 | 94.40 | 26.22 | 54.0 | 95.14 | 26.20 |
| + ResFields ($i=1,2,3$) | 46.1 | 94.81 | 27.00 | 44.6 | 95.23 | 27.20 | 49.6 | 93.92 | 27.66 | 48.1 | 94.81 | 26.80 | 42.3 | 95.28 | 26.32 |
| + ResFields ($i=1,\dots,7$) | 47.6 | 95.03 | 27.16 | 49.6 | 95.42 | 27.51 | 46.3 | 94.29 | 27.96 | 43.5 | 94.96 | 26.74 | 50.9 | 95.43 | 26.44 |
| DyNeRF (Li et al., 2022) | 60.8 | 93.68 | 25.78 | 60.9 | 94.03 | 25.89 | 56.2 | 92.60 | 26.16 | 70.1 | 93.46 | 25.26 | 56.1 | 94.63 | 25.79 |
| + ResFields ($i=1$) | 52.6 | 94.25 | 26.42 | 55.9 | 94.59 | 26.70 | 55.5 | 93.23 | 27.08 | 50.5 | 94.22 | 26.13 | 48.4 | 94.95 | 25.75 |
| + ResFields ($i=1,2,3$) | 51.8 | 94.42 | 26.53 | 51.3 | 94.94 | 26.74 | 55.3 | 93.31 | 27.07 | 48.8 | 94.40 | 26.33 | 51.6 | 95.06 | 25.99 |
| + ResFields ($i=1,\dots,7$) | 48.9 | 94.60 | 26.72 | 51.5 | 95.04 | 26.95 | 55.7 | 93.63 | 27.40 | 45.7 | 94.50 | 26.23 | 42.8 | 95.22 | 26.28 |
| DNeRF | 138.3 | 92.54 | 24.40 | 128.8 | 92.90 | 24.34 | 92.1 | 91.19 | 25.04 | 191.5 | 92.33 | 23.50 | 140.7 | 93.76 | 24.72 |
| + ResFields ($i=1$) | 56.5 | 94.36 | 26.47 | 48.0 | 95.05 | 27.03 | 63.8 | 92.89 | 26.35 | 61.9 | 94.29 | 26.09 | 52.2 | 95.19 | 26.41 |
| + ResFields ($i=1,2,3$) | 49.6 | 94.59 | 26.68 | 49.3 | 95.23 | 27.14 | 64.0 | 93.07 | 26.58 | 42.2 | 94.87 | 26.75 | 42.8 | 95.18 | 26.22 |
| + ResFields ($i=1,\dots,7$) | 52.7 | 94.81 | 26.88 | 47.5 | 95.32 | 26.90 | 61.6 | 93.44 | 26.98 | 49.0 | 94.99 | 26.87 | 52.5 | 95.47 | 26.77 |
| Nerfies (Park et al., 2021a) | 135.0 | 93.57 | 25.35 | 97.9 | 93.55 | 25.21 | 135.5 | 93.10 | 26.20 | 186.5 | 93.41 | 24.73 | 120.1 | 94.23 | 25.26 |
| + ResFields ($i=1$) | 52.0 | 94.75 | 26.99 | 48.0 | 95.16 | 27.14 | 51.7 | 93.96 | 27.79 | 55.3 | 94.59 | 26.36 | 53.1 | 95.29 | 26.66 |
| + ResFields ($i=1,2,3$) | 45.3 | 94.80 | 26.88 | 41.7 | 95.18 | 26.70 | 50.4 | 93.99 | 27.82 | 42.7 | 94.72 | 26.49 | 46.4 | 95.31 | 26.52 |
| + ResFields ($i=1,\dots,7$) | 42.2 | 94.90 | 26.73 | 51.8 | 95.31 | 26.71 | 23.6 | 93.84 | 26.97 | 43.6 | 95.06 | 26.76 | 49.8 | 95.41 | 26.49 |
| HyperNeRF (Park et al., 2021b) | 63.5 | 94.67 | 26.51 | 59.9 | 94.64 | 26.01 | 57.8 | 94.25 | 27.55 | 69.7 | 94.44 | 25.75 | 66.7 | 95.35 | 26.74 |
| + ResFields ($i=1$) | 46.9 | 94.69 | 26.80 | 41.0 | 95.15 | 27.14 | 50.4 | 93.83 | 27.70 | 45.6 | 94.73 | 26.54 | 50.7 | 95.04 | 25.80 |
| + ResFields ($i=1,2,3$) | 47.1 | 94.85 | 26.99 | 40.7 | 95.40 | 27.41 | 46.5 | 93.90 | 27.71 | 49.2 | 94.89 | 26.75 | 52.0 | 95.20 | 26.07 |
| + ResFields ($i=1,\dots,7$) | 48.0 | 95.07 | 27.27 | 50.7 | 95.46 | 27.50 | 49.5 | 94.14 | 27.90 | 44.8 | 95.22 | 27.03 | 46.9 | 95.46 | 26.66 |
| NDR (Cai et al., 2022) | 66.2 | 94.50 | 26.48 | 64.1 | 94.84 | 26.64 | 55.8 | 94.05 | 27.40 | 78.1 | 93.93 | 25.34 | 66.8 | 95.17 | 26.53 |
| + ResFields ($i=1$) | 49.5 | 94.71 | 26.89 | 50.4 | 95.02 | 26.90 | 51.8 | 93.85 | 27.64 | 47.6 | 94.67 | 26.48 | 48.3 | 95.32 | 26.53 |
| + ResFields ($i=1,2,3$) | 47.5 | 94.89 | 27.13 | 46.2 | 95.36 | 27.36 | 51.0 | 93.86 | 27.62 | 46.4 | 94.91 | 26.85 | 46.1 | 95.42 | 26.69 |
| + ResFields ($i=1,\dots,7$) | 49.8 | 94.97 | 27.08 | 44.2 | 95.31 | 27.14 | 50.7 | 94.20 | 27.82 | 49.3 | 95.03 | 26.87 | 55.2 | 95.37 | 26.52 |

Table A.8: **Capacity of the low-rank representation.** We measure the capacity of the low-rank parametrization on 351x510-resolution videos (250 frames) with varying levels of difficulty: *1)* a video composed of randomly selected images (250 segments), *2)* a video consisting of 6 coherent segments, and *3)* a full video depicting one sequence. We observe that increasing the number of independent segments has a significant effect on the model performance due to the lack of information that is shared across the entire signal and could be encoded in the base matrix weights. In spite of this, our method successfully improves quality over the vanilla Siren even in the most challenging case (**29.18** *vs.* **19.4** PSNR for rank 40).

| | Rank | Mean test PSNR↑ | train PSNR↑ | Random Video (250 segments) test PSNR↑ | train PSNR↑ | Bikes Video (6 segments) test PSNR↑ | train PSNR↑ | Cat Video (1 segment) test PSNR↑ | train PSNR↑ |
|---|---|---|---|---|---|---|---|---|---|
| Siren | | 29.86 | 30.21 | 19.40 | 19.65 | 33.68 | 34.05 | 36.49 | 36.94 |
| +ResFields | 1 | 32.99 | 33.52 | 23.34 | 23.78 | 36.85 | 37.40 | 38.77 | 39.38 |
| | 2 | 34.18 | 34.89 | 24.26 | 24.89 | 38.26 | 38.98 | 40.02 | 40.80 |
| | 4 | 35.83 | 36.82 | 25.51 | 26.47 | 39.95 | 40.94 | 42.02 | 43.06 |
| | 8 | 37.31 | 38.73 | 26.84 | 28.39 | 41.46 | 42.82 | 43.62 | 45.00 |
| | 10 | 37.73 | 39.33 | 27.26 | 29.09 | 41.86 | 43.34 | 44.07 | 45.57 |
| | 20 | 38.74 | 41.08 | 28.39 | 31.50 | 42.82 | 44.77 | 45.01 | 46.96 |
| | 40 | 39.43 | 42.54 | 29.18 | 33.64 | 43.32 | 45.85 | 45.78 | 48.12 |

## A.6 ADDITIONAL INSIGHTS

**Capacity of the low-rank representation.** Learning signals with many independent parts imposes additional challenges for both ResFields and coordinate MLPs in general. The coordinate MLP weights possess the ability to compress a spatio-temporal signal into a compact representation, mainly because the source signal has a significant amount of repetitive information which is effectively represented by the optimized network weights. The shared weights of the ResFields have the same property, as they tend to store information that is common across the entire signal, whereas the low-rank weights accommodate for topological and dynamic changes for effective learning.

We conduct the following experiment to analyze the learning capacity of our low-rank weights in the case of dynamic scenes with varying complexity. We create three videos with different levels of difficulty: 1) A corner case when every frame depicts a novel scene (a video composed from the first

250 images from the DIV2K dataset Agustsson & Timofte (2017)), 2) the *bikes* video containing 6 segments, and 3) the cat video containing only 1 segment. All of these three videos are trimmed and cropped to the same length (250 frames) and resolution ($351 \times 510$). Then, we perform the 2D video approximation (analogous to Sec. 4.1) by learning the mapping from a space-time coordinate to RGB color (10% of pixels are not seen during the training and are left for evaluation).

Results are reported in Tab. A.8. We observe that increasing the number of independent video segments indeed has a significant effect on the model performance. This can be attributed to the lack of information that can be shared across the entire signal and efficiently encoded in the base matrix weights. However, despite this, our method successfully improves quality over the vanilla Siren, even in the most challenging case of a random video (19.4 vs 29.18 PSNR for our model with rank 40). This improvement can be attributed to (a) the increased capacity of the pipeline thanks to residual modeling and (b) the ability of the neural network to discover small patterns that could be effectively compressed into shared weights even in the extreme case of a video with completely random frames. The latter behavior is especially apparent in the simplest case when ResFields uses only rank 1 (19.4 vs 23.34 PSNR for the random video sequence). Please also note that increasing the capacity in our case does not lead to significant computational overhead, as we demonstrate in the main submission and further emphasize in the next section.

**Modeling long sequences.** ResFields approach faces limitations for long and evolving signals, as the shared weight matrix runs out of capacity. Here, a straightforward strategy to deal with longer sequences utilized in the literature (Li et al., 2022) is to split the sequence into independent chunks and train separate neural fields. We investigate different strategies for addressing the sequences of arbitrary length.

As our test bed, we consider the 2D video approximation task (Sec. 4.1) with a longer 512x288-resolution video (1024 frames). Here, we re-purpose the video captured by one of our Kinects for the dynamic NeRF reconstruction task. We divide the video sequence into several chunks of varying sizes: 512, 256, 128, and 64 frames (2, 4, 8, and 16 sub-sequences respectively). We consider maintaining for each chunk a set of shared $\mathbf{W}_i$ and residual weights $\boldsymbol{\mathcal{W}}_i(t)$ as well as both together $(\mathbf{W}_i, \boldsymbol{\mathcal{W}}_i(t))$. For example, having 4 chunks and both shared and delta weights updated would mean having 4 separate tensors and 4 separate factorized delta weights. The results are summarized in the Tab. A.9.

Table A.9: **Modeling long sequences.**

| | #params [M] | Chunking #chunks | part | PSNR↑ test | train |
|---|---|---|---|---|---|
| Siren | 0.2 | 1 | | 28.18 | 28.21 |
| +ResFields | 2.2 | 1 | | 34.58 | 34.78 |
| | **2.6** | | shared | 35.65 | 35.87 |
| | 4.2 | 2 | residual | 35.96 | 36.22 |
| | 4.6 | | both | **37.01** | 37.31 |
| | **3.0** | | shared | 37.06 | 37.32 |
| | 8.1 | 4 | residual | 37.30 | 37.66 |
| | 8.9 | | both | **38.89** | 39.31 |
| | **3.8** | | shared | 38.58 | 38.90 |
| | 16.0 | 8 | residual | 38.36 | 38.84 |
| | 17.5 | | both | **40.49** | 41.08 |
| | **5.3** | | shared | 39.55 | 39.96 |
| | 31.7 | 16 | residual | 39.47 | 40.06 |
| | 34.8 | | both | **41.32** | 42.13 |

We observe that as we increase the number of sub-sequences, the results gradually improve for all three strategies, with the best overall quality achieved by independently updating both weights. However, this strategy naturally requires the largest amount of parameters. We noted that the strategy of chunking only shared weights is the most parameter-efficient for processing coherent long sequences. Specifically, maintaining 16 shared weights for the whole sequence achieves better quality compared to maintaining 4 sets of all considered network parameters (39.55 *vs.* 38.89 test PSNR), while using fewer parameters (5.3M *vs.* 8.9M).

