# OpenReview forum: "ResFields: Residual Neural Fields for Spatiotemporal Signals"
_ICLR.cc/2024/Conference — ICLR 2024 spotlight_

### Official Review · Reviewer_KYty · 2023-10-30

**Soundness:** 4 excellent
**Presentation:** 4 excellent
**Contribution:** 3 good
**Rating:** 8
**Confidence:** 4

**Summary:**

The paper proposes a method for improving the performance of coordinate-based network representation of various temporally varying signals. The key contribution of the method is that rather than having the network learn to represent temporal variation in a completely unstructured way, the temporal variation is instead modeled by a low-rank weight matrix which is added to a base weight matrix that is shared across all time instances. This exploits the temporal consistency in the underlying signals being represented, and results in a more parameter efficient and high-quality representation of the underlying signal. This is benchmarked across various tasks in which coordinate-based networks are used, such as video overfitting, dynamic SDF fitting, and dynamic neural radiance field fitting, where ResFields (the contribution) demonstrate consistent increased performance.

**Strengths:**

In my opinion the strengths of the paper are as follows:
1. The paper is described exceptionally clearly, and the method makes intuitive sense on why it would improve performance for representation on temporal signals. The relative simplicity of the method and lack of reliance on a number of complex hyperparameters makes it more likely that future methods will be influenced and use the contributions.
2. The evaluations are thorough, across three different temporal signal representation tasks, all demonstrating clear improvements from the ResField contribution. In each of these domains, a number of baseline methods are all compared against. I found that the comparisons on dynamic NeRF especially are thorough, using many different state-of-the-art methods and all showing that ResFields leads to increased performance.
3. The method is ablated well, and various ablations show the contribution of the method is actual source of the improvements.

**Weaknesses:**

In my opinion, the main weaknesses of the paper are:
1. To me, it is unclear how much capacity the low-rank representation has to model various dynamic components. While this may not be a problem in something like dynamic NeRF or dynamic SDF where the signal remains relatively constant over time, for overfitting a 2D video, where there may be cuts or completely different scenes, I would be interested to see how the method performs. In this case, it seems like the low-rank weight matrix approximation would be required to model all of the signal, as there is little consistency shared in the underlying signal over time and thus less information to store in the shared weights. Could this potentially be a limitation of the method?
2. One additional minor weakness is that the method seems like it may add significant amount of computational overhead. For every iteration, the weight matrix of the underlying representation needs to be changed. Does this result in significant slowdown in optimization? It would be an insightful comparison to include this in the baseline comparisons, especially in the dynamic NeRF scenario where speed of fitting is very important. If this adds a significant amount of computational overhead, it is possible that the amount of gains is not worth it, and could be achieved by simply training a standard representation for longer.

**Questions:**

I do not have any additional questions on the manuscript. Overall, as described in the strengths section, I see the paper as a good paper: the method is described well, simple, and makes intuitive sense, and then it is evaluated well across a number of tasks and ablated to show that the improvement comes from the proposed contribution. For this reason I am positive on the paper and leaning towards acceptance. I see some minor weaknesses in the potential capacity of the model, and especially in the potential computational overhead of the method in training speed. Addressing these weaknesses would incline me to raise my scores for the paper.

**Update after author response**

Thank you for the detailed response. It has addressed my questions. I have updated my score in accordance.

---

> ### Author Response · Authors · 2023-11-14
> **Related to the capacity of the low-rank representation (weakness 1/2)**
>
> We would like to thank the reviewer for considering our work as exceptionally clearly described and thoroughly evaluated and ablated, as well as for the insightful comments and questions regarding the proposed method and its performance. In the following, we address the remaining discussion points related to the capacity of the low-rank representation and the potential computational overhead of the proposed framework.
>
>  ### 1. The capacity of the low-rank representation for modeling dynamic components of varying complexity.
> We especially appreciate the suggestion and we will incorporate an additional study to benchmark the learning capabilities of the low-rank weights for signals with multiple diverse dynamic segments.
>
>
> Learning signals with many independent parts imposes additional challenges for both ResFields and coordinate MLPs in general. The coordinate MLP weights possess the ability to compress a spatio-temporal signal into a compact representation, mainly because the source signal has a significant amount of repetitive information which is effectively represented by the optimized network weights. The shared weights of the ResFields have the same property, as they tend to store information that is common across the entire signal, whereas the low-rank weights accommodate for topological and dynamic changes for effective learning.
>
>
> We conduct the following experiment to analyze the learning capacity of our low-rank weights in the case of dynamic scenes with varying complexity. We create three videos with different levels of difficulty: 1) A corner case when every frame depicts a novel scene (a video composed from the first 250 images from the DIV2K dataset), 2) the bikes video containing 6 segments, and 3) the cat video containing only 1 segment. All of these three videos are trimmed and cropped to the same length (250 frames) and resolution ( 351x510 ).
>
>
> Then, we perform the 2D video approximation (analogous to Sec. 4.1.) by learning the mapping from a space-time coordinate to RGB color (10% of pixels are not seen during the training and are left for evaluation).
>
> |  |     | Mean | | Random Video  | (250 segments) | Bikes Video  | (6 segments) | Cat Video  | (1 segment) |
> | :------------- |:--:|:--:|:--:|:--:|:--:|:--:|:--:|:--:|:--:|
> |  | Rank     | test PSNR  | train PSNR | test PSNR  | train PSNR | test PSNR  | train PSNR | test PSNR  | train PSNR |
> Siren |  | 29.86 | 30.21 | 19.40 | 19.65 | 33.68 | 34.05 | 36.49 | 36.94 |
>  +ResFields | 1 | 32.99 | 33.52 | 23.34 | 23.78 | 36.85 | 37.40 | 38.77 | 39.38 |
>  +ResFields | 2 | 34.18 | 34.89 | 24.26 | 24.89 | 38.26 | 38.98 | 40.02 | 40.80 |
>  +ResFields | 4 | 35.83 | 36.82 | 25.51 | 26.47 | 39.95 | 40.94 | 42.02 | 43.06 |
>  +ResFields | 8 | 37.31 | 38.73 | 26.84 | 28.39 | 41.46 | 42.82 | 43.62 | 45.00 |
>  +ResFields | 10 | 37.73 | 39.33 | 27.26 | 29.09 | 41.86 | 43.34 | 44.07 | 45.57 |
>  +ResFields | 20 | 38.74 | 41.08 | 28.39 | 31.50 | 42.82 | 44.77 | 45.01 | 46.96 |
>  +ResFields | 40 | 39.43 | 42.54 | 29.18 | 33.64 | 43.32 | 45.85 | 45.78 | 48.12 |
>
>
> Generally, we observe that increasing the number of independent video segments indeed has a significant effect on the model performance. This can be attributed to the lack of information that can be shared across the entire signal and efficiently encoded in the base matrix weights. However, despite this, our method successfully improves quality over the vanilla Siren, even in the most challenging case of a random video (19.4 vs 29.18 PSNR for our model with rank 40). This improvement can be attributed to (a) the increased capacity of the pipeline thanks to residual modeling and (b) the ability of the neural network to discover small patterns that could be effectively compressed into shared weights even in the extreme case of a video with completely random frames. The latter behavior is especially apparent in the simplest case when ResFields uses only rank 1 (19.4 vs 23.34 PSNR for the random video sequence). Please also note that increasing the capacity in our case does not lead to significant computational overhead, as we demonstrate in the main submission and further emphasize in the next section.

---

> > ### Author Response · Authors · 2023-11-14
> > **Related to the computational overhead of the ResFields model (weakness 2/2)**
> >
> > ### 2. Computational overhead of the ResFields model.
> >
> > Please note that in many cases, ResFields in fact speeds up the training time, as it enables the usage of smaller MLPs. Specifically, as we showed in the appendix for the video approximation task  (Table A.5), Siren+ResFields with 512 neurons and Siren with 1700 neurons have an equal number of parameters and demonstrate similar performance. However, optimizing a huge MLP with 1700 neurons comes with a much greater computational cost. Our method achieves over **six times** faster training while requiring **over two times less GPU memory**.
> >
> > Purely integrating ResFields into an existing MLP architecture (e.g. in dynamic NeRFs) has a negligible computational overhead on training and inference speed, while bringing considerable improvement in terms of quality. The inference time of NeRF models is almost not affected by the incorporation of the ResFields block (see the FPS column in Tab. 3), since calculating weight matrices for a particular time step is lightweight and extremely fast.
> >
> > The training time of NeRFs with ResFields (i=1,...7) models is slower by only ~1% (runtime measured on the "Dancer" Owlii sequence). Specifically, the training step of TNeRF baseline takes 9.37it/s while TNeRF+ResFields (on 7 layers) takes 9.28it/s (11.86 vs 11.97 hours of training).
> > For other MLP baselines, the relative computational overhead is even smaller percentage-wise, as they rely on additional modules that do not involve ResFields. E.g. HyperNeRF vs HyperNeRF+ResFields(i=1,...7) is 3.51it/s vs 3.49it/s (31.65h vs 31.84h).
> >
> > Additionally, the weight matrices are not accessed randomly during either training or inference, they are batched according to the respective time frame IDs and processed in parallel with a marginal computational overhead, as supported by the reported runtimes.
> >
> > We will include the training time for each NeRF method in Tab. 3. to provide a more comprehensive overview on the computational overhead and further showcase the advantage of the proposed residual weight modeling.
> >
> > Lastly, we would like to thank the reviewer again for their time and effort. In the revised manuscript, we will incorporate the additional quantitative and qualitative results, as well as necessary clarifications for the aforementioned experiments related to the raised discussion points. We believe that this additional evaluation will provide further evidence for the efficiency and practicality of our method. We would be glad to hear further comments and discussion on the potential directions of improvement for our manuscript.

---

> ### Author Response · Authors · 2023-11-22
> **Follow up**
>
> Dear Reviewer KYty,
>
> We would like to kindly inquire whether you have had the opportunity to review our updated manuscript, along with the responses addressing your raised discussion points. Your feedback is highly valuable to us, and we welcome any additional questions or comments you may have.
>
> Best regards,
>
> Authors

---

### Official Review · Reviewer_ZKNQ · 2023-11-01

**Soundness:** 3 good
**Presentation:** 3 good
**Contribution:** 3 good
**Rating:** 8
**Confidence:** 4

**Summary:**

This paper propose a way to extend MLP-based neural fields to have the capability to model time-dependent signals. The key ideas is to modulate the residual of the MLP weights by a time-dependent matrices decomposed into a vector matrix. This decomposition share a right basis, which allows reusing structures between time steps, providing regularity and potentially helping with generality. The paper shows good quantitative results that support the claim. I believe this idea is simple and effective, and it’s can be a good contribution for the community.

**Strengths:**

- The idea is very simple. The simplicity of such idea allows it to be combined with different design of neural fields as long as the main training parameters are parameterized by matrices.
- The quantitative results shows that it’s also effective to certain degree. The ablation studies also seems to support the effectiveness of techniques.

**Weaknesses:**

- (minor) The paper provides little intuition of why this particular factorization is chosen rather than some alternative ways to factorize these matrices. For example, can we make v(t) to be N_ixR_i and M_i to be R_ixM_i? Is there any intuitive reason why this is not a good choice? Maybe this is addressed in the ablation study in Section 4.4, but providing some more intuition is good.
- (minor) While the idea seems to support any matrix-like weight parameterization, the specific factorization might provide different interpretation depending on the way the network use the weight matrix. Maybe the specific design choice is limited to MLP architecture.
- Concern about long sequence. The sequence weight at time t is modeled as W_o + \Delta W_t. When t is very far from the original frame, then the weights \Delta W_t can have significant weights, so it’s not clear to me whether sharing the same matrix M across all time steps is a good idea. Maybe the author can consider periodically updating the weight W_o once a while?

**Questions:**

See weakness section.

---

> ### Author Response · Authors · 2023-11-15
> **Alternative weight matrix factorization, key intuition (weaknesses 1/3)**
>
> We thank the reviewer for considering our work simple and effective, and that it can be a good contribution to the community. In the following, we address the remaining discussion points:
>
> ### 1. Alternative weight matrix factorization, key intuition.
>
> We have explored diverse modeling and factorization techniques (Tab. 5, 6) based on prior work, including different tensor factorizations (CP, Tucker), however, we have not considered the proposed example. We will gladly include it in our ablation study (Tab. 6).
>
> The goal of our parametrization is to achieve high learning capacity while retaining good generalization properties. To achieve this for a limited budget in terms of the number of parameters, we want to allocate as few as possible independent parameters per time step (represented by the vector $v(t)$) and as many globally shared parameters $M_i$. Allocating more capacity towards the shared weights will enable _1)_ the increased capacity of the model due to its ability to discover small patterns that could be effectively compressed in shared weights and _2)_ stronger generalization as the model is aware of the whole sequence through the shared weights.
>
> Given these two objectives, we have designed $v(t)$ to have very few parameters (R) and $M_i$ to have the most parameters (RxNxM).
>
> In light of this intuition, the factorization proposed by the reviewer appears to be suboptimal, as it generally allocates more capacity in the per-frame independent part $v(t)$ (NxR), and less into the shared weights $M_i$ (RxM).
>
> We further verify this intuition quantitatively and show that the alternative design has a lower learning capacity, due to a larger number of independent parameters that need to be re-learned for every time step; for the same reason, this parametrization is more prone to overfitting. Below, we provide the extended Tab. 6 on the video approximation task:
> | Factorization | $v(t)$ |     $M_i$ | Rank | #params | test PSNR | train PSNR |
> | :-------------|:----|:----|:----:|:-------:|:---------:|:----------:|
> | proposed      |${N_i \times R_i}$ | ${R_i \times N_i}$          | 10   | 5.4     |     35.22  | 36.35     |
> | proposed      |${N_i \times R_i}$ | ${R_i \times N_i}$ | 20   | 10.0    |     35.88  | 37.50     |
> | proposed      |${N_i \times R_i}$ | ${R_i \times N_i}$  | 40   | 19.3    |     36.67  | 39.01     |
> |
> | ours          |${R_i}$| ${R_i \times N_i \times M_i }$| 10   | 8.7     |     39.59  | 40.80     |
> | ours          |${R_i}$| ${R_i \times N_i \times M_i }$| 20   | 16.5     |     40.87 | 42.45     |
>
> Quantitative results confirm that our parametrization achieves higher learning capacity (higher training PSNR) and stronger generalization (higher test PSNR) while using fewer parameters.
>
> Specifically, our parametrization with the rank of 10 (8.7M parameters) compared to the proposed example with the rank of 40 (19.3M parameters) demonstrates:
>
> - test PSNR: **39.59** vs 36.67
> - train PSNR: **40.8** vs 39.01
> - the number of parameters: **8.7M** vs 19.3M
>
> We will include these results and further elaborate on our intuition in the revisited manuscript and the supplementary material.

---

> > ### Author Response · Authors · 2023-11-15
> > **Limitation to MLPs (weakness 2/3)**
> >
> > We study the representation of spatio-temporal signals through the lenses of neural fields. Hence, we focus solely on the MLP architecture as a dominant parametrization utilized for this task. However, incorporating ResFields into different neural network architectures is an interesting avenue for future work.

---

> > > ### Author Response · Authors · 2023-11-15
> > > **Modeling longer sequences (weakness 3/3)**
> > >
> > > ### 3. Modeling longer sequences.
> > >
> > > ResFields approach can indeed face limitations for long and evolving signals, as the shared weight matrix runs out of capacity. Here, a straightforward strategy to deal with longer sequences utilized in the literature [1] is to split the sequence into independent chunks and train separate neural fields.
> > >
> > > Updating the base weight matrix, as suggested by the reviewer, is a valuable alternative idea for overcoming the aforementioned natural limitation. To investigate different strategies for addressing the sequences of arbitrary length, we will include the following study in the revised manuscript and appendix.
> > >
> > >
> > >
> > > As our test bed, we consider the 2D video approximation task (Sec. 4.1) with a longer 512x288-resolution video (1024 frames). Here, we repurpose the video captured by one of our Kinects for the dynamic NeRF reconstruction task (Section 4.3).
> > >
> > > We divide the video sequence into several chunks of varying sizes: 512, 256, 128, and 64 frames (2, 4, 8, and 16 sub-sequences respectively). We consider maintaining for each chunk a set of shared $W_O$ and residual $\Delta W_t$ weights as well as both together ($W_O$ and $\Delta W_t$). E.g., having 4 chunks and both shared $W_O$ and delta weights $\Delta W_t$ updated would mean having 4 separate tensors $W_O$ and 4 separate delta weights factorizations.
> > >
> > > The results are summarized in the Table below.
> > > |        | #params [M] | #chunks | chunking part | test PSNR | train PSNR |
> > > | :------|:------------:|:-------:|:-------------:|:---------:|:----------:|
> > > Siren    | 0.2 | 1 |  | 28.18 | 28.21 |
> > > +ResFields | 2.2 | 1 |  | 34.58 | 34.78 |
> > > |
> > > +ResFields | **2.6** | 2 | shared ($W_O$) | 35.65 | 35.87 |
> > > +ResFields | 4.2 | 2 | residual ($\Delta W_t$) | 35.96 | 36.22 |
> > > +ResFields | 4.6 | 2 | both | **37.01** | 37.31 |
> > > |
> > > +ResFields | **3.0** | 4 | shared ($W_O$) | 37.06 | 37.32 |
> > > +ResFields | 8.1 | 4 | residual ($\Delta W_t$) | 37.30 | 37.66 |
> > > +ResFields | 8.9 | 4 | both | **38.89** | 39.31 |
> > > |
> > > +ResFields | **3.8** | 8 | shared ($W_O$) | 38.58 | 38.90 |
> > > +ResFields | 16.0 |8  | residual ($\Delta W_t$) | 38.36 | 38.84 |
> > > +ResFields | 17.5 |8  | both | **40.49** | 41.08 |
> > > |
> > > +ResFields | **5.3** | 16 | shared ($W_O$) | 39.55 | 39.96 |
> > > +ResFields | 31.7 |16  | residual ($\Delta W_t$) | 39.47 | 40.06 |
> > > +ResFields | 34.8 |16  | both | **41.32** | 42.13 |
> > >
> > >
> > > We observe that as we increase the number of sub-sequences, the results gradually improve for all three strategies, with the best overall quality achieved by independently updating both weights. However, this strategy naturally requires the largest amount of parameters.
> > >
> > > We noted that the strategy suggested by the reviewer is the most parameter-efficient for processing coherent long sequences. Specifically, maintaining 16 shared weights $W_O$ for the whole sequence achieves better quality compared to maintaining 4 sets of all considered network parameters (39.55 vs 38.89 test PSNR), while using fewer parameters (5.3M vs 8.9M).
> > >
> > > We thank the reviewer again for this valuable idea; the provided analysis will be included in the revised manuscript and appendix.
> > >
> > > We welcome additional comments and discussions on potential ways to improve our manuscript.
> > >
> > >
> > >
> > >
> > >    [1] Li, Tianye, et al. "Neural 3d video synthesis from multi-view video." Proceedings of the IEEE/CVF Conference on Computer Vision and Pattern Recognition. 2022.

---

> ### Author Response · Authors · 2023-11-22
> **Follow up**
>
> Dear Reviewer ZKNQ,
>
> We would like to kindly inquire whether you have had the opportunity to review our updated manuscript, along with the responses addressing your raised discussion points. Your feedback is highly valuable to us, and we welcome any additional questions or comments you may have.
>
> Best regards,
> Authors

---

### Official Review · Reviewer_4JwP · 2023-11-01

**Soundness:** 4 excellent
**Presentation:** 4 excellent
**Contribution:** 3 good
**Rating:** 8
**Confidence:** 4

**Summary:**

This paper addresses the challenge of modeling non-static scenes with NeRF, where introducing a temporal dimension 't' significantly increases the required model capacity. A method for temporal information modeling is proposed, which by employing a low-rank representation, controls the amount of parameters to be learned. This approach enables temporal modeling without affecting the network size. Alongside, this paper innovates by integrating temporal residual layers in neural fields, dubbed ResFields, showcasing an effective way to represent complex temporal signals without increasing the size of Multi-Layer Perceptrons (MLPs), thus offering a promising solution to the capacity bottleneck for modeling and reconstructing spatiotemporal signals.

**Strengths:**

Strengths:
The method proposed in this paper is simple and effective, with clear and easy-to-understand writing, and ample experiments.
Novelty:
This paper introduces a lightweight, plug-and-play module to enhance NeRF's  capability for dynamic objects.
Generalization:
This paper applies ResFields to various base models and downstream tasks, achieving a certain performance improvement across the board.

**Weaknesses:**

The main issue is that this paper lacks references to and comparisons with closely related methods, such as [1][2][3][4]. These works all address the problem of modeling non-static scenes, which is closely related to the theme of this paper. These methods primarily model a normalized space, then model the relationship between the 3D expression at each moment 't' and the normalized space. The absence of this comparison leads to (1) unclear performance advantages, and (2) uncertainty about whether ResFields could also be applied to these methods to further enhance temporal modeling capability.

**Questions:**

1. The symbols in Equation 5 are not explained.
2. Why are the results of NGP plus ResFields not shown in Table 1?
3. Can ResFields, like LoRA[5], provide further dynamic modeling capabilities to a pre-trained NeRF?

[1]Wang C, MacDonald L E, Jeni L A, et al. Flow supervision for Deformable NeRF[C]//Proceedings of the IEEE/CVF Conference on Computer Vision and Pattern Recognition. 2023: 21128-21137.
[2]Li Z, Wang Q, Cole F, et al. Dynibar: Neural dynamic image-based rendering[C]//Proceedings of the IEEE/CVF Conference on Computer Vision and Pattern Recognition. 2023: 4273-4284.
[3] Wang Y, Han Q, Habermann M, et al. Neus2: Fast learning of neural implicit surfaces for multi-view reconstruction[C]//Proceedings of the IEEE/CVF International Conference on Computer Vision. 2023: 3295-3306.
[4] Wang Q, Chang Y Y, Cai R, et al. Tracking Everything Everywhere All at Once[J]. arXiv preprint arXiv:2306.05422, 2023.
[5]Hu E J, Shen Y, Wallis P, et al. Lora: Low-rank adaptation of large language models[J]. arXiv preprint arXiv:2106.09685, 2021.

---

> ### Author Response · Authors · 2023-11-21
> **Comparisons to flow-supervised NeRF methods [1,2] (Weakness 1/2)**
>
> We would like to thank the reviewer for considering our work effective and easy to use, as well as for the insightful comments and references. In the following, we do our best to put the presented ResFields approach in the context of the provided related methods. We provide the quantitative evaluation when possible, and elaborate on the conceptual differences and limitations when a straightforward comparison is not feasible.
>
> ### Comparisons to flow-supervised NeRF methods [1,2]
>
> Please note that both methods [1,2] require optical flow supervision, which in practice is hard to obtain for scenes considered in the paper. The Owlii sequences exhibit a magnitude of uniformly colored texture patterns that has a significant impact on the optical-flow estimators (e.g. RAFT).
>
> Unfortunately, [1] has no published code at the present day, significantly limiting our ability to provide quantitative evaluation against the approach in a timely manner.
>
> We have evaluated DynIBR [2] on the Owlii benchmark (Tab. 3). However, the method fully fails on the dataset with large camera view angle differences. This behavior can be partially attributed to the reliance of the method on the synthesized depth maps obtained from the off-the-shelf monocular depth regressors, which produce inconsistent results across largely varying camera views. Additionally, DynIBR being an image-based rendering technique in its nature, also exhibits a rather limited novel-view synthesis and geometry reconstruction capability and serves more as a video stabilization technique. Finally, training DynIBR (and its variation with ResFields) is not computationally feasible for us as the original method already requires 8 A100 GPUs for training.
>
> However, in the following we provide strong evidence that ResFields approach generally benefits the task of modeling _motion fields_, the fundamental building block of the aforementioned flow-based NeRF methods [1,2].
>
> Specifically, DynIBR [2] uses a 256-neuron ReLU-MLP to model bi-directional motion fields by predicting coefficients of the DCT bases, while Deformable NeRF [1] uses a deformation network of Nerfies (128-neuron ReLU-MLP) to predict motion as SE(3). In both cases, the NeRF models are explicitly supervised with optical flow to learn motion/scene flow.
>
> To demonstrate the benefit of ResFields for both of these motion fields, we use 4 sequences of DeformingThings from Sec. 4.2, and learn bi-directional motion fields. We use 80% of tracked mesh vertices to learn the motion field and the remaining 20% for evaluation. As a supervision, we use L1 error between the predicted and the ground truth motion.
>
> The table below reports the L1 error (scaled by x10^3) for the evaluation points. Additionally, we have included one additional modeling option from [A], where the motion field is simply predicted as an offset vector.
>
>
> |                        | it/s | flow type | fwd/bwd $l_1$ error |
> | :----------------------|:----:|:---------:|:-------------------:|
> | ReLU-MLP (128 neurons) |  **15.6** | offset | 6.88 / 7.31 |
> | + ResFields |  **15.6** | offset | **3.85 / 3.85** |
> |
> | ReLU-MLP (128 neurons) |  **15.6** | SE(3) | 4.57 / 4.58 |
> | + ResFields |  **15.6** | SE(3) | **2.64 / 2.56** |
> |
> | ReLU-MLP (128 neurons) |  **15.6** | DCT | 3.19 / 3.19 |
> | + ResFields            |  **15.6** | DCT | **2.18 / 2.19** |
> |
> |
> | ReLU-MLP (256 neurons) |  5.6 | offset | 6.50 / 6.44 |
> | + ResFields            |  5.6 | offset | **4.43 / 4.59** |
> |
> | ReLU-MLP (256 neurons) |  5.6 | SE(3) | 4.00 / 4.10 |
> | + ResFields            |  5.6 | SE(3) | **2.88 / 2.84** |
> |
> | ReLU-MLP (256 neurons) |  5.6 | DCT | 2.47 / 2.48 |
> | + ResFields            |  5.6 | DCT | **1.79 / 1.79** |
>
>
> Note that ResFields greatly benefits learning motion fields across all settings. Moreover, the particular architecture from DynIBR (DCT with 256 neurons) is less powerful than its much smaller and faster counterpart (128 neurons) with ResFields for both forward (2.18 vs 2.47) and backward (2.19 vs 2.48) motion while being around **three times faster** (16.5 vs 5.6 training it/s).
>
> References:
>
> [A]: Prokudin et. al., “Dynamic Point Fields” ICCV 2023

---

> > ### Author Response · Authors · 2023-11-21
> > **Additional comparisons with concurrent methods [3,4] (Weakness 2/2)**
> >
> > Please note that both NeuS2 [3] and Tracking Everything [4] are concurrent submissions (ICCV Oct. 2023).
> >
> > However, in the following table we compare NeuS2 [3] to the proposed ResField approach on the Owlii dataset (Tab. 3).
> > |        | _Mean_ CD | _Mean_ SSIM | _Mean_ PSNR |
> > | :------|:------------:|:-------:|:-------------:|
> > | NeuS2   |  69.4 | 91.01 | 21.73 |
> > | Tensor4D|  32.8 | 91.05 | 22.59 |
> > | TNeRF   |  17.2 | 94.18 | 26.18 |
> > | TNeRF+ResFields (i=1,...,7)|  **14.2** | **95.45** | **27.55** |
> >
> > It can be seen that ResFields consistently surpasses the performance of the NeuS2 on all metrics by a large margin. This can be attributed to the nature of the NeuS2 approach, which represents each time step with a separate set of parameters, only using the model from a previous time frame for initialization. This approach fails considerably in the case of the limited number of views (four in the case of our Owlii setup) available for each time step. Please also note that representing dynamic scenes in such a manner results in excessive data storage, as each time step scene takes approximately 25 Mb of data. For comparison, storing a single dynamic sequence of 100 frames requires 250 Mb for NeuS2 and 5 Mbs for our ResField model (TNeRF+ResFields).
> >
> > Tracking Everything Everywhere All at Once [4] is addressing a specific problem of point tracking in 2D videos, and is not directly applicable to any of the tasks considered in the present submission. However, it is indeed related when viewed from a broader perspective of dynamic 3D consistent scene modeling via implicit neural fields. Here, this work shares considerable similarities to the Neural Surface Reconstruction (NDR) method in terms of (a) how the deformation of the canonical space is defined (invertible neural network) and (b) how it is used to query a neural field in the canonical space. We have demonstrated in Tab. 3 that our ResFields benefits and is compatible with NDR.
> >
> > We will add a detailed discussion and the aforementioned quantitative results of the baselines in the revised version of the manuscript and its supplementary.

---

> > > ### Author Response · Authors · 2023-11-21
> > > **Answers to additional questions**
> > >
> > > ## Symbols in Eq. 5.  (Question 1/3)
> > >
> > > We thank the reviewer for the suggestion on improving the clarity of the method description.
> > >
> > > Following [B], all models are supervised by minimizing the pixel-wise difference between the rendered and ground truth colors ($l_1$ loss), the rendered opacity and the gt mask (binary cross-entropy), and further adopting the Eikonal mean squared error loss for well-behaved surface reconstruction under the sparse capture setup:
> > >
> > > $ L = L_{color} + \lambda_1 L_{igr} + \lambda_2 L_{mask},$
> > >
> > > where $L_{color}$  is the $l_1$ loss between the rendered and the ground truth value, $L_{igr}$ is the mean squared error between the magnitude of a unit vector and the corresponding predicted normal, and $L_{mask}$ is the binary cross entropy loss between the rendered opacity and the gt mask. The hyperparameters ($\lambda_1$,$\lambda_2$) are set to 0.1 as specified in the appendix.
> > >
> > >
> > > References:
> > >
> > > [B] Wang, Peng, et al. "Neus: Learning neural implicit surfaces by volume rendering for multi-view reconstruction." NeurIPS (2021).
> > >
> > > ## Compatibility with NGP. (Question 2/3)
> > > The main goal of the ResFields approach is to improve the ability of MLPs to represent complex spatiotemporal signals as neural network weights. However, in the case of instant NGP, the signal is represented mostly in the multi-resolution hash grid structure, and the utilized small MLP is mostly used for blending stored hash features. Storing the data in the hash grid structure effectively resolves the problem of limited MLP capacity, alleviating the need for residual weights. This, however, comes at the expense of the decreased generalization capability, as we show in the original Figure 4 and Table 1 (video approximation experiment).
> > >
> > > Empirically, we find that representing the small blending MLP of instant NGP via ResFields brings almost no improvement in terms of the reconstruction quality (+0.02 improvement in PSNR), verifying our conjecture regarding the high capacity (yet poor generalization) of the instant NGP. We will emphasize this discussion in the revised version of the submission.
> > >
> > > ## ResFields for efficient fine-tuning. (Question 3/3)
> > > While both ResFields and LoRA utilize low-rank parametrization of neural network weight matrices, the main premises of the two methods are different. The main idea of LoRA is to efficiently fine-tune a pre-trained model by leveraging a low-rank parametrization of a large weight matrix of the baseline model. In the case of ResFields, introducing the residual weights in a low-rank parametrized way is aimed at increasing the capacity and expressive power of the baseline model. The two tasks appear to be complementary, with no apparent way to utilize ResFields for the fine-tuning task.
> > >
> > > We would like to thank the reviewer again for their time and effort. In the revised manuscript, we will incorporate the aforementioned discussion and additional results. Furthermore, a dedicated paragraph will be incorporated to specifically address the flow-based neural field methods, providing a more contextual understanding of recent approaches and their relevance to our proposed methodology. We welcome any further suggestions to enhance the overall quality of the manuscript.

---

> ### Comment · Reviewer_4JwP · 2023-11-22
>
> The rebuttal addresses my concerns well. I will raise the score.

---

### Author Response · Authors · 2023-11-22

We would like to thank all reviewers for their constructive feedback and insightful comments.
We especially appreciate that reviewers consider our work simple (_R 4JwP_, _R ZKNQ_, _R KYty_), clearly written (_R 4JwP_, _R KYty_), and thoroughly evaluated (_R KYty_).

We have responded to the comments of each reviewer and subsequently revised both our manuscript and the supplementary appendix, as outlined below.
1. We included a comparison with an additional concurrent work NeuS2 (_R 4JwP_) and provided training times for each method in Tab. 3 (_R KYty_) to highlight the efficiency of our method.
2. (_R 4JwP_) We compared scene flow models and their variations with ResFields (Sec. 4.4)
3. (_R 4JwP_) We clarified symbols in Eq. 5 and discussed compatibility with NGP (Sec. 4)
4. (_R ZKNQ_) We included an additional factorization technique in Tab. 7 and elaborated on our key idea (Sec. 3)
5. (_R ZKNQ_) We provided an additional study on how to process longer sequences (A. 6)
6. (_R KYty_) We analyzed the capacity of the low-rank representation and included an additional ablation study (A. 6)

We sincerely thank our reviewers and look forward to further discussions.

---

### Meta-Review · Area_Chair_aKLy · 2023-12-11

**Metareview:**

The paper proposes a time-dependent layer for MLPs for reconstructing spatiotemporal signals.

All three reviewers are positive about this work. It's simple and effective, lightweight and plug-and-play module.

Some remaining concerns are:
- missing discussions/comparisons with several closely related work,
- concern about long sequences. unclear if the low-rank representation has sufficient capacity to model the dynamics.

During the rebuttal, the authors provided detailed responses, revised the paper with additional comparisons, and clarified the key intuition of the paper.

The AC believes that this is a clear accept.

**Justification For Why Not Higher Score:**

The proposed method is compatible only with MLP-based neural fields. This may limit the applicability since most of the recent state-of-the-art neural fields are hybrid/explicit (e.g., multi-res hashgrid, or 3D Gaussian).

**Justification For Why Not Lower Score:**

The paper introduces a simple yet effective approach. The AC belives that it deserves a spotlight presentation.

---

### Decision · Program_Chairs · 2024-01-16

Accept (spotlight)